# *StatsMerging*: Statistics-Guided Model Merging via Task-Specific Teacher Distillation

## Abstract

As large models are increasingly deployed across various tasks, the limited GPU memory available for storing and executing task-specific models presents a growing bottleneck. Model merging has emerged as a promising solution to accommodate multiple large models within constrained memory budgets. While traditional multi-task learning methods attempt to merge common layers, they require labor-intensive annotated labels and incur significant computational overhead. Recent merging techniques aim to address this issue by combining models at inference time; however, these approaches often rely on simplistic heuristics, ignore weight distribution characteristics, assume architectural identity, or require access to test samples to infer merging coefficients, thereby limiting generalization and scalability. We present *StatsMerging*, a novel lightweight learning-based model merging method guided by weight distribution statistics without requiring ground truth labels or test samples. *StatsMerging* offers three key advantages: (1) It uniquely leverages singular values from singular value decomposition (SVD) to capture task-specific weight distributions, serving as a proxy for task importance to guide task coefficient learning; (2) It employs a lightweight learner *StatsMergeLearner* to model the weight distributions of task-specific pre-trained models, improving generalization and enhancing adaptation to unseen samples; (3) It introduces *Task-Specific Teacher Distillation* for merging vision models with heterogeneous architectures, a merging training paradigm that avoids costly ground-truth labels by task-specific teacher distillation. Notably, we present two types of knowledge distillation, (a) distilling knowledge from task-specific models to train *StatsMergeLearner*; and (b) for the first time, distilling knowledge from models with different architectures prior to merging, following a distill-then-merge paradigm. Extensive experiments across vision and NLP tasks demonstrate the effectiveness of *StatsMerging*. Our results show that *StatsMerging* outperforms state-of-the-art techniques, achieving overall accuracies of $94.5\%$ for Vision and $77.6\%$ for NLP, while further exhibiting strong generalization to unseen tasks, and robustness to image quality variations.

## 1 Introduction

Computer vision has witnessed transformative progress fueled by deep learning, particularly through the development and adoption of large-scale pre-trained models. Architectures like Convolutional Neural Networks (CNNs) (Krizhevsky et al., 2012; He et al., 2016; Simonyan & Zisserman, 2014), Vision Transformers (ViTs) (Dosovitskiy et al., 2021b; Touvron et al., 2021), and hybrid approaches (Liu et al., 2022b) pre-trained on massive datasets have become the cornerstone of modern vision applications. Large-scale models leveraging multi-modal pre-training, such as CLIP (Radford et al., 2021)) or generative models like GANs (Goodfellow et al., 2014) and Diffusion Models (Ho et al., 2020; Rombach et al., 2022) have further pushed the boundaries of visual understanding and synthesis, enabling the use of pre-trained backbones to a wide range of downstream vision applications. The dominant practice is to fine-tune these powerful base models to computer vision tasks, including image classification (He et al., 2016), object detection (Ren et al., 2015; Carion et al., 2020a), semantic segmentation (Long et al., 2015; Xie et al., 2021), image restoration (Zhang et al., 2017; Saharia et al., 2022), and image generation (Mirza & Osindero, 2014). This success, however, leads to a practical challenge: the proliferation of numerous specialized pre-trained weights and model checkpoints (Cao et al., 2024a; 2025), most of which share the same foundational ViT or CNN backbones.

Managing this growing collection incurs significant storage overhead, complicates deployment, and represents a missed opportunity to consolidate the related, yet specialized, knowledge contained within these models (Wortsman et al., 2022), particularly on compute-constrained platforms such as edge devices (Cao et al., 2024b; Singh et al., 2024). While Multi-Task Learning (MTL) (Vandenhende et al., 2022b) aims to create versatile single models for vision tasks, it often demands complex joint training strategies, concurrent access to diverse datasets, and careful architecture design to balance performance across disparate tasks.

Model merging offers a compelling post-hoc alternative, seeking to combine independently trained models without expensive retraining. However, while techniques for model merging have gained traction, particularly in Natural Language Processing (NLP) (Yadav et al., 2023a; Ilharco et al., 2023), adapting these techniques to tasks in computer vision domain has been far less explored. A straightforward approach of simple weight averaging (Wortsman et al., 2022) often fails in vision tasks due to the complex, hierarchical visual feature representations, task-specific optimizations, and the presence of intricate noise patterns that lead to sharp, non-convex loss minima (Izmailov et al., 2018). Recent methods in this direction (Matena & Raffel, 2022; Jin et al., 2023; Yang et al., 2023; Padmanabhan et al., 2023) neglect the importance of weight distribution.

This paper introduces a novel model merging framework specifically designed to address the afore-mentioned challenges, for computer vision as well as NLP tasks. We propose *StatsMerging*, a weight distribution statistics-guided merging approach that moves beyond simple parameter averaging or task-vector manipulation. *StatsMerging* leverages the statistical features from models pre-trained on prior tasks. We compute salient statistics extracted by leveraging Singular Value Decomposition (SVD) to capture the dominant properties of the learned feature spaces. This statistical information, intrinsically captures aspects of the pre-trained model distributions and guides the merging process by learning a compact Multilayer Perceptron (MLP), coined *StatsMergeLearner* that predicts adaptive merging coefficients ($\lambda$) shown in Fig. 1.

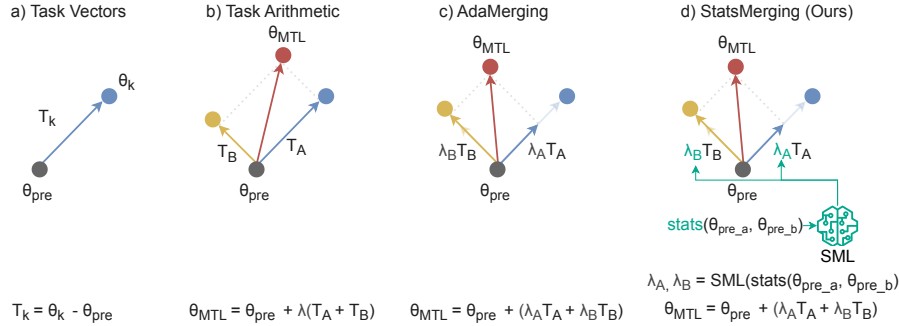

Figure 1: Compared to prior works, *StatsMergeLearner* uniquely learns the merging coefficients by exploiting statistical features of weights pre-trained on prior tasks. Notably, while both AdaMerging and *StatsMerging* are presented in the task-wise level in c) and d) for simplicity of illustration, the same principle can be applied at the layer-wise level for fine-grained adaptation.

We make four significant contributions summarized as follows:

- We propose *StatsMerging*[1], a novel model merging framework guided by model weight statistics, leveraging SVD to predict merging coefficients $\lambda$.

- We design the lightweight *StatsMergeLearner* to learn model merging coefficients $\lambda$ estimation based on statistical features of model weights, through a newly proposed Task-Specific Teacher Distillation training paradigm without manually-annotated labels.

- We introduce the first heterogeneous architectural merging method, which distills knowledge from models with non-identical architectures into the unified target architecture.

- Extensive experiments demonstrate the effectiveness of our proposed *StatsMerging* for model merging, achieving state-of-the-art average accuracies of $94.5\%$ on Vision tasks and $77.6\%$ on NLP tasks.

---

[1]Our code is available at https://github.com/statsmerging/statsmerging.

## 2 RELATED WORK

**Multi-Task Learning.** Multi-Task Learning (MTL) (Zhang & Yang, 2021; Vandenhende et al., 2022a) represents a paradigm for training a single model to perform multiple tasks concurrently. While MTL aims to create unified models capable of handling diverse objectives, it typically requires careful design of network architectures, computationally expensive training, access to large and diverse datasets, and intricate task balancing strategies (Zhang & Yang, 2021). Furthermore, MTL necessitates joint training from the outset, which can be computationally expensive and may not be feasible when dealing with a collection of pre-trained, specialized models. Model merging offers a compelling alternative by enabling the combination of independently trained models, without the need for extensive retraining or simultaneous access to multi-task datasets.

**Multi-Task Merging.** Early approaches to model merging often involved simple heuristics like Weight Averaging (Wortsman et al., 2022), TIES-Merging (Yadav et al., 2023a), and Arithmetic Merging (Ilharco et al., 2023). While straightforward to implement, these methods (Ye et al., 2023; Akiba et al., 2025; Tang et al., 2025) typically lack awareness of the weight distributions and learned representations within the models, leading to suboptimal performance in the merged model compared to individually fine-tuned models or unified models trained from scratch. For instance, naive weight averaging could significantly degrade performance (Wortsman et al., 2022), highlighting the challenges in consolidating knowledge from independently trained networks. Recent work decomposes models into common and task-specific subspaces to achieve isotropic merging (Marczak et al., 2025). Task Singular Vectors (TSV) (Gargiulo et al., 2025) is proposed to reduce interference among tasks by aligning merging operations along task-relevant directions. Methods in NLP (Yadav et al., 2023b; Ilharco et al., 2023) have shown promise by learning interpolation weights.

**Statistical Characterization for Model Merging.** Prior work examines statistical patterns in fine-tuned models, but typically relies on these signals individually. Foundational second-order analyses show that task-specific learning induces shifts in weight means and variance (LeCun et al., 1989; Hassibi & Stork, 1992), both serve as lightweight, data-free approximation to Fisher information (Kirkpatrick et al., 2017; Matena & Raffel, 2022). However, such gradient-based merging methods require costly and task-independent computation. Magnitude-based importance has been studied extensively in pruning and sparse sub-networks (Frankle & Carbin; Molchanov et al., 2019; Zhu & Gupta, 2017), and in model patching frameworks that identify high-magnitude, task-relevant components (Goel et al.). A complementary line of work shows that fine-tuning updates concentrate in a small number of dominant SVD directions, revealing strong low-rank structure (Ilharco et al., 2023; 2022; Ortiz-Jimenez et al., 2023), consistent with findings for model merging from task arithmetic (Ilharco et al., 2023), model soups (Wortsman et al., 2022), transformer transferability (Narang et al., 2021), intrinsic dimensionally (Li et al., 2018), and neural anisotropy (Ortiz-Jiménez et al., 2020). Similarly, factorization-based knowledge distillation leverages low-rank decompositions to transfer structured task information (Liu et al., 2022a). However, these approaches either depend on expensive gradients or isolate only one statistical feature of mean, variance, magnitude or low-rank structure. Through a lightweight *StatsMergeLearner*, our work combines mean, variance-as-Fisher signals, magnitude, and dominant SVD directions to jointly capture task structure for efficient, label-free model merging.

| Method | No Manual Label | No TT Samples | Layer Level | TT Adaptability | Heterogeneous Architecture |
|---|---|---|---|---|---|
| Traditional MTL | ✗ | ✗ | * | ✗ | ✗ |
| Task Arithmetic | ✓ | ✓ | ✗ | ✗ | ✗ |
| TIES-Merging | ✓ | ✓ | ✗ | ✓ | ✗ |
| Fisher Merging | ✓ | ✓ | ✗ | ✗ | ✗ |
| RegMean | ✓ | ✓ | ✗ | ✗ | ✗ |
| AdaMerging | ✓ | ✗ | ✓ | ✓ | ✗ |
| *StatsMerging* (Ours) | ✓ | ✓ | ✓ | ✓ | ✓ |

Table 1: Summary of system characteristics in recent works. *: Optional. TT: Test-Time. Test-time adaptability refers to the ability of a model to adjust its weights to unseen data during inference without access to human-labeled annotations..

In summary, our method *StatsMerging* enjoys several advantages compared to prior works: (1) it eliminates the need of human annotated labels; (2) remains lightweight with marginal overhead; (3) is explicitly designed to support heterogeneous architectures; and (4) provides flexibility for test-time adaptability summarized in Table 1.

## 3 METHODOLOGY

### 3.1 PRELIMINARIES

**Notations:** A deep neural network is parameterized by a set of weights $\theta = \{\theta_1, \theta_2, \ldots, \theta_L\}$ that learns the mapping from an input data $x_i \in \mathbb{R}^d$ to a predicted value $\hat{y}_i \in \mathbb{R}^D$: $f_\theta(x_i) \to \hat{y}_i$. Of these, $\theta^l$ represents the $l$-th $l \in \{1, 2, \ldots, L\}$ layer weights where $L$ is the number of layers of the model $f_\theta$, $d$ denotes an input data $x_i$'s dimension. For classification problems, $y_i$ is the class label and $D$ is the number of classes, while for regression problems, $D$ is the dimension of the output vector $y_i$.

The weights of a pre-trained model (e.g., ViT or ResNet) are denoted as $\theta_{pre} = \{\theta^1_{pre}, \theta^2_{pre}, \ldots, \theta^L_{pre}\}$. The weights fine-tuned on a specific training data $\{x_i, y_i\}_{i=1}^{N_k^{tr}}$ for task $k$ is recorded as $\theta_k = \{\theta^1_k, \theta^2_k, \ldots, \theta^L_k\}$ where $N_k^{tr}$ is the number of training samples.

**Problem Formulation:** The problem of *model merging* is formulated as: given $K$ tasks' training data, find a way to combine weights $\{\theta_k\}_{k=1}^K$ fine-tuned for $K$ tasks previously to obtain a new weight $\theta_m$ without undergoing the retraining process, while the new model $f_{\theta_m}$ is capable of performing well on $K$ tasks jointly.

It is assumed that all $K$ fine-tuned weights and the merged weight share the same neural network architecture. Therefore, the core question is how to *linearly combine* $\{\theta_k\}_{k=1}^K$ to obtain $\theta_m$. In the task level, the model merging problem is finding a set of coefficients $\lambda_k \in \{\lambda_1, \lambda_2, \ldots, \lambda_K\}$ such that the merged model weights $\theta_m = \sum_{k=1}^K \lambda_k \theta_k$ for model $f_{\theta_m}$ perform well on all $K$ tasks. At the layer level, it becomes searching for a set of coefficients $\lambda_k^l \in \{\lambda_1^1, \lambda_1^2, \ldots, \lambda_1^L, \lambda_2^1, \lambda_2^2, \ldots, \lambda_2^L, \ldots, \lambda_K^L\}$ to obtain the merged model $\theta_m = \sum_{k=1}^K \sum_{l=1}^L \lambda_k^l \theta_k^l$ that maintain high performance on $K$ tasks.

### 3.2 WEIGHT STATISTICS-GUIDED MODEL MERGING

In this section, we describe the main intuition and techniques of our proposed method: *StatsMerging*. **Motivation:** Fisher-based methods estimate parameter importance through second-order sensitivity (Kirkpatrick et al., 2017; Matena & Raffel, 2022) that represents local per-parameter importance (Amari, 1998; Kunstner et al., 2019), requiring explicit costly gradient computation. Prior studies highlight signals such as magnitude (Frankle & Carbin; Molchanov et al., 2019; Goel et al.) or low-dimensional task directions (Ilharco et al., 2023; 2022; Ortiz-Jimenez et al., 2023), each revealing structured effects of fine-tuning but typically treated in isolation. Inspired by these insights, we adopt the design principle of jointly leveraging simple, data-free statistics including mean and variance as lightweight Fisher proxies [2] with additional global information, magnitude, and dominant SVD components to capture complementary facets of task structure for efficient, label-free model merging.

Building on these insights, we use weight statistics as compact representations of the weight distribution, avoiding raw weights which are prohibitively high-dimensional. These summarized distributions of pre-trained weights $\theta_k$ enable the prediction of merging coefficients through a function $g(\theta_k) \to \lambda_m$. The resulting statistics encode task-relevant information about how each model $\theta_k$ contributes to the final merged model.

**Weight Statistics:** For a pre-trained weight $\theta_k$ on task $k$, we compute the mean $\mu_{\theta_k}$ and variance $\sigma^2 = Var(\theta_k)$ to represent its center and breadth, as well as its magnitude $m = ||\theta_k||$. In addition, we extract the singular values $\sigma'_i$ from Singular Value Decomposition (SVD):

$$W_k = U_k \Sigma_k V_k^\top \tag{1}$$

where $W_{\theta_k}$ represents the matrix of the model parameter $\theta_k$. By default, we use rank 3 from $\Sigma_k$ to form weight statistics. We hypothesize that singular values compress the key information regarding

---

[2]See Sec. A.4.2 for the derivation.

weight distribution that can benefit the decision of assigning the amount of weights from $\theta_k$ for merging. Combining all together, the weight statistics feature vector $S_k$ is formed as

$$S_k = stats(\theta_k) = [\mu, \sigma^2, m, \sigma'_r] \tag{2}$$

where $stats()$ extracts the statistical features from the weight $\theta_k$, $\sigma_r$ represents the singular value vector given rank $r$: $\boldsymbol{\sigma'_r} = [\sigma'_1, \sigma'_2, \ldots, \sigma'_r]$. Our empirical results indicate that a rank 3 approximation is effective in extracting key weight information.

Notably, the Equation 3 above is task-wise while we also introduce layer-wise formulation for layer $l$:

$$S_k^l = stats(\theta_k^l) = [\mu, \sigma^2, m, \sigma'_r]^l \tag{3}$$

where the layer-wise statistics features of pre-trained model from task $k$ layer $l$ is computed.

**StatsMergeLearner (SML):**  We adopt a multilayer perceptron (MLPs) to learn to predict the merging coefficients $\lambda$ given weight statistics feature vector $S_k$ as input. In the task-wise mode, the *StatsMergeLearner* is denoted as $SML(S_k)$:

$$\lambda_k = SML(S_k) = g(stats(\theta_k)) \tag{4}$$

where $\lambda_k$ is a scalar representing the merging coefficient of Task $k$ model. In the layer-wise mode, the *StatsMergeLearner* is denoted as $M(S_k)$:

$$\lambda_k^l = SML(S_k^l) = g(stats(\theta_k^l)) \tag{5}$$

where $\lambda_k$ is a vector containing $L$ layers' coefficients and $\lambda_k^l$ refers to the coefficient of layer $l$ in the $k$ pre-trained model. *StatsMerging* is carefully designed that a simple two-layer MLP which serves as the default learner, is sufficient to learn effective model merging coefficients, as demonstrated in Section A.5.4.

**Optimization Objective.** To train *StatsMergeLearner* (SML), in the standard supervised training paradigm, we freeze the weights for each task $\theta_k$ and apply the cross-entropy loss function $L_{CE}$ on the aggregated dataset:

$$\mathcal{L}_{\text{CE}}^{SL} = -\sum_{c=1}^{C_m} y_c \log(\hat{y}_c) \tag{6}$$

where $\hat{y}_c$ is the prediction from the merged model for class $c$, $C_m$ is the total number of classes in the aggregated dataset [3].

### 3.3 TASK-SPECIFIC TEACHER DISTILLATION

The requirement of labeled data for training SML can pose a significant burden, as aggregating labels across $K$ tasks incurs substantial cost. This challenge is further exacerbated when the labels must be manually annotated by humans. Such high costs further hinder the broader applicability of SML. We ask the following **research question**: *Is there a feasible way to obtain sufficiently reliable labels for effective SML learning without incurring the labor-intensive costs of manual annotation?*

Observe that, in the model merging context, $K$ pre-trained models are already given. With the help of well-trained teachers, knowledge distillation (Hinton et al., 2015) has been proven as an effective way to train a model without human annotations. Therefore, when aggregating samples from $K$ tasks together with their respective task experts (depicted as gurus in the figures), high-quality labels can be obtained at *no additional manual cost*.

These observations guide our design of a novel Task-Specific Teacher Distillation paradigm that trains the *StatsMergeLearner* (SML) for model merging. We illustrate the overview in Fig. 2 and detailed in Algorithm 1. The intermediate process of pseudo label generation and the role of pseudo labels are further depicted in Fig 3 (a) and (b), respectively.

---

[3]The theoretical analysis is provided in Sec. A.4.

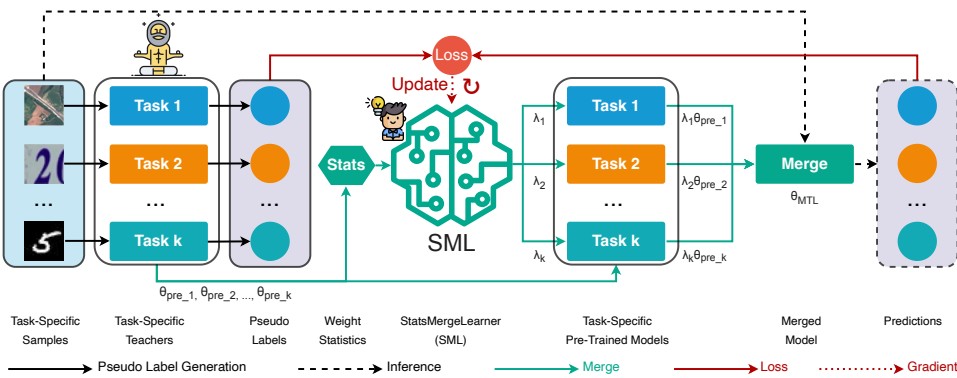

Figure 2: *StatsMerging* Overview. *StatsMergeLearner* (SML) learns the merging coefficients $\lambda$ by minimizing the loss between the merged model's predictions and pseudo labels generated by task-specific teachers. During inference, only the merged model is employed to predict class labels.

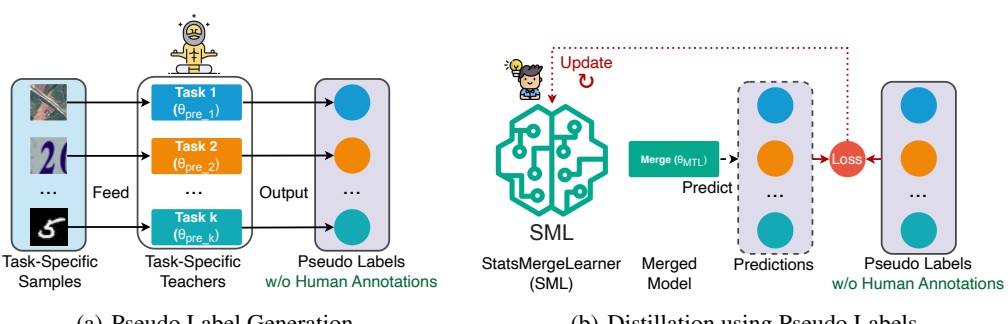

(a) Pseudo Label Generation          (b) Distillation using Pseudo Labels

Figure 3: Depiction of Task-Specific Teacher Distillation procedure. (a) Pseudo labels are generated by feeding samples into Task-Specific Teachers; (b) depicts the roll of distillation labels: the discrepancy between the predictions from the merged model and the pseudo labels from (a) is computed through the loss function, further update StatsMergLearner's parameters. w/o: without.

The key intuition behind the Task-Specific Teacher Distillation is that each pre-trained model $\theta_k$ already performs well on its own task dataset where $\{x_i, y_i\}_k \in D_k$. We regard it ($\theta_k$) as the Task-Specific Teacher $T_k$. Subsequently, the predictions $\hat{y}_{i,k}$ from the model trained on task $k$ are sufficiently reliable to serve as high-quality pseudo labels for the corresponding pre-trained dataset sample $\{x_i, y_i\}_k$. We aggregate such pairs $\{x_i, \hat{y}_i\}_k$ to construct the merged dataset to train SML. The key benefit of this approach is that it enables dataset preparation without relying on human-annotated labels. The predicted class label in one-hot encoded format. Therefore, the cross-entropy loss is applied while such loss function simplicity helps extend to other tasks and architectures in vision and NLP domain:

$$\mathcal{L}_{\text{CE}} = -\sum_{c=1}^{C_m} \hat{y}_{c,k} \log(\hat{y}_c)). \tag{7}$$

Algorithm 1. Unified Statistics-Guided Model Merging via Task-Specific Teacher Model Distillation[a]

1: **Input:** Set of pre-trained models $\{M_1, M_2, \ldots, M_k\}$ with weights $\{\theta_1, \theta_2, \ldots, \theta_k\}$ for $K$ tasks.
2: **Output:** Merged model $M_{\text{merged}}$ with weights $\theta_{\text{merged}}$
3: // Prepare $K$ pre-trained models
4: **if** Same architecture $A$ for all $M_i$ **then**
5:     Set $M_{\text{target}}$ to the shared architecture
6: **else**
7:     Select a target architecture $M_{\text{target}}$
8:     **for** $i = 1$ to $k$ **do**
9:         **if** $A(M_i) \neq A(M_{\text{target}})$ **then**
10:           Distill $M_i$ into $M_{\text{target}}$ to obtain updated $\theta_i$
11:         **end if**
12:     **end for**
13: **end if**
14: // Merge $K$ models
15: **for** $k = 1$ to $K$ **do**
16:     // mean $\mu$, std $\sigma^2$, norm $m$, singular value $\sigma'_r$
17:     Extract statistics $S_k = [\mu, \sigma^2, m, \sigma'_r]$ from $\theta_k$
18:     Predict coefficients $\lambda_k = \text{SML}(S_k)$
19:     Merge layer weights: $\theta^l_{\text{merged}} = \sum_{i=1}^k \lambda_k \theta_k$
20: **end for**
21: **return** $M_{\text{merged}}$ with weights $\theta_{\text{merged}}$

[a]Distillation is detailed in Appendix A.3

## 4 EXPERIMENTS AND EVALUATION

### 4.1 EXPERIMENTAL SETUP

In this section, we present the experimental setup and evaluation results used to compare our method against recent baselines.

**Datasets and Models**: Our experiments include eight image classification tasks with datasets SUN397 (Xiao et al., 2016), Stanford Cars (Krause et al., 2013), RESISC45 (Cheng et al., 2017), EuroSAT (Helber et al., 2019), SVHN (Netzer et al., 2011), GTSRB (Stallkamp et al., 2011), MNIST (LeCun et al., 1998), DTD (Cimpoi et al., 2014), and CIFAR10 (Krizhevsky, 2009) [4] We use ViT-B/32 CLIP (Radford et al., 2021) as the pre-trained backbone. Individual task-specific models are obtained by training on each dataset separately. For merging models with different architectures, we first distill them into a single backbone before applying our merging method.

**Baselines and Metrics**: We compare against standard baselines including Individual Training, Traditional Multi-Task Learning (MTL) (Zhang & Yang, 2021), Weight Averaging (Wortsman et al., 2022), Task Arithmetic (Ilharco et al., 2023), Fisher Merging (Matena & Raffel, 2022), RegMean (Jin et al., 2023), TIES-Merging (Yadav et al., 2023a) and AdaMerging (Yang et al., 2023). The primary evaluation metric is the average accuracy (Avg Acc) on the test sets of all tasks. The evaluation is conducted on eight different vision classification tasks.

*StatsMergeLearner* **Training Detail**: Our MLP-based *StatsMergeLearner* learns to predict layer-wise or task-wise merging weights coefficients ($\lambda$) based on weight statistics from individual task models. The *StatsMergeLearner* is trained for 500 epochs using Adam, with a learning rate of $1e-3$ and a StepLR scheduler (factor 0.1 every 100 epochs), which translates to around only 3 hours to merge 4 ViTs, offering the practicality and advantage of applying our technique for practitioners without spending days or weeks for training (Zhang & Yang, 2021; Padmanabhan et al., 2023). We train the *StatsMergeLearner* primarily using knowledge distillation from the aggregated dataset without human annotated labels described in Sec. 3.3, optimized with either Cross-Entropy (Mao et al., 2023) or KL Divergence (Kullback & Leibler, 1951) loss.

### 4.2 MERGING PERFORMANCE

In this section, we present a comprehensive evaluation of our approach in comparison to state-of-the-art task vector merging methods, assessing its superiority across several fundamental aspects: Multi-task merging performance, generalization to unseen tasks and heterogeneous architectures.

**Improved Merging Performance.** Our proposed framework *StatsMerging* demonstrated state-of-the-art (SOTA) performance spanning eight **vision** and seven **NLP** tasks, shown in Fig. 4 across various model **scales** [5].

In **Vision** tasks, *StatsMerging* achieved 84.5% (ViT-B/32) and 92.1% (ViT-L/14) average accuracy (Avg Acc). With 40% more available validation samples, *StatsMerging++* further improved to 94.5% (B, +10.0%) and 94.1% (L, +2.0%), outperforming WEMoE (84.5%, 93.6%) and AdaMerging (81.1%, 91.0%). We attribute the improvements to the ability of *StatsMergeLearner* to adapt task-specific weights based on their weight statistics to the merged model. The use of pseudo labels from task-specific teachers provides stronger signals for *StatsMergeLearner* in assigning weight coefficients compared to AdaMerging entropy minimization and more complex task-adaptive expert selection mechanism in WEMoE.

On **NLP** benchmarks, *StatsMerging* reached 77.6% (T5 Base) and 77.5% (T5 Large) Avg Acc, surpassing the second best method TIES-Merging (Val) of 73.9% (+3.7%) and 74.4% (+3.1%).

---

[4]In the remainder of the paper, the abbreviations shown in brackets are used to denote each task dataset: Vison tasks – SUN397 (SU), Cars (CA), RESISC45 (RE), EuroSAT (EU), SVHN (SV), GTSRB (GT), MNIST (MN), and DTD (DT); NLP tasks – PAWS (PA), QASC (QA), QuaRTz (QR), Story Cloze (SC), WikiQA (WQ), Winogrande (WG) and WSC (WS).

[5]Please refer to the Appendix for experimental details, including the full list of tasks, datasets, baselines, along with the task-level results in Sections A.1 and A.2, respectively.

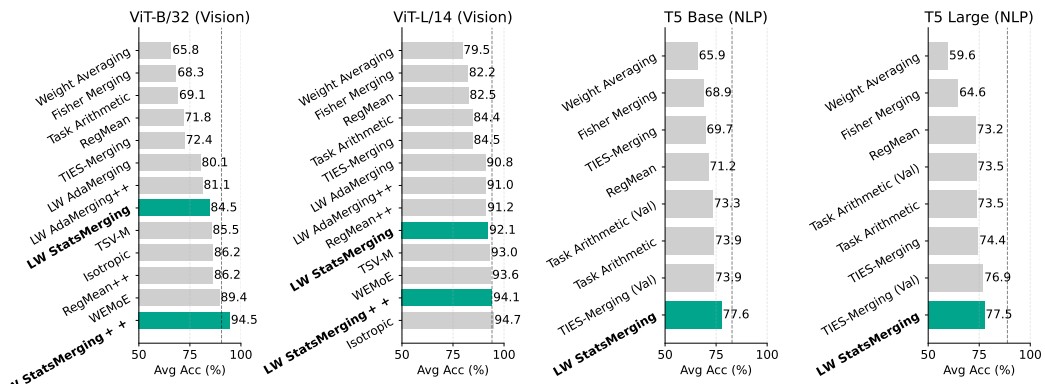

Figure 4: LW *StatsMerging++* achieved state-of-the-art performance on eight **Vision** and seven **NLP** tasks across various scales, highlighted in teal-green in the figures. Each number represents the average accuracy (Avg Acc) across tasks. *StatsMerging++* further improved *StatsMerging* by scaling validation input samples. The performance of each individual fine-tuned model is shown as dashed vertical reference lines.

**Marginal Parameter and Computation Overhead.** SML is lightweight in terms of parameters and computation. Our 2-Layer *StatsMergeLearner* with the merged model contain 10.99M parameters, requires 2.95 GFLOPs, and achieves an inference time of 5.26 ms on an NVIDIA RTX A6000 GPU.

Without the merged model, *StatsMergeLearner* (SML) itself is orders of magnitude smaller and computationally lighter than the merged model, with only 0.336M parameters, 0.73M MACs and 1.46M FLOPs. The results demonstrate that SML introduces negligible overhead in terms of parameters (SML-to-Merged Model Parameter Ratio: 0.336M / 10.99M = 0.0306) and computation (SML-to-Merged Model Compute Ratio: 1.46M / 2.95G = 0.0005).

**Significantly Enhanced Generalization.** A merged model is expected to generalize to unseen tasks by strategically transferring the knowledge from the combined set of old tasks. We benchmarked such generalization ability of *StatsMerging* against four strong baselines: Task Arithmetic, TIES-Merging, AdaMerging, and AdaMerging++. We followed the same evaluation protocol in AdaMerging training on two groups of tasks, each group consisting of six seen tasks, and testing on two unseen tasks.

Table 2: Generalization results (Avg Acc %) on two unseen tasks when merging Layer-Wise ViT-B/32 models on six tasks. *StatsMerging*: shaded in gray. Bold: top score. Underscore: 2nd-highest score.

| | **Seen Tasks** | | | | | | | **Unseen Tasks** | | |
|---|---|---|---|---|---|---|---|---|---|---|
| **Method** | SU | CA | RE | DT | SV | GT | **Avg Acc** | MN | EU | **Avg Acc** |
| Task Arithmetic | 63.3 | 62.4 | 75.1 | 57.8 | 84.6 | 80.4 | 70.6 | 77.2 | 46.2 | 61.7 |
| TIES-Merging | 67.8 | 66.2 | 77.2 | 56.7 | 77.1 | 70.9 | 69.3 | 75.9 | 43.3 | 59.6 |
| AdaMerging | 65.2 | 65.9 | **88.5** | 61.1 | 92.2 | 91.5 | 77.4 | 84.0 | 56.1 | 70.0 |
| AdaMerging++ | 68.2 | 67.6 | 86.3 | 63.6 | 92.6 | 89.8 | 78.0 | 83.9 | 53.5 | 68.7 |
| *StatsMerging* | **69.1** | **71.3** | 86.7 | **75.2** | **93.2** | **95.7** | **81.9 (+3.9)** | **85.1** | **56.4** | **70.8 (+0.8)** |

| **Method** | SU | CA | GT | EU | DT | MN | **Avg Acc** | RE | SV | **Avg Acc** |
|---|---|---|---|---|---|---|---|---|---|---|
| Task Arithmetic | 64.0 | 64.0 | 75.2 | 87.7 | 57.0 | 95.7 | 73.9 | 52.3 | 44.9 | 51.1 |
| TIES-Merging | 68.0 | 67.1 | 67.7 | 78.4 | 56.5 | 92.8 | 71.8 | **58.7** | 49.2 | 53.9 |
| AdaMerging | 67.1 | 67.8 | 94.8 | 94.4 | 59.6 | 98.2 | 80.3 | 50.2 | 60.9 | 55.5 |
| AdaMerging++ | 68.9 | 69.6 | 91.6 | 94.3 | 61.9 | 98.7 | 80.8 | 52.0 | 64.9 | 58.5 |
| *StatsMerging* | **69.6** | **73.3** | **96.1** | **95.4** | **74.1** | **97.2** | **84.3 (+3.5)** | 54.2 | **67.1** | **60.7 (+2.2)** |

Details are presented in Table 2, where in both groups our proposed *StatsMerging* achieved 70.8% and 60.7%, significantly outperforming the second best method AdaMerging by +0.8% and +2.2% margins. Such improvements are attributed to both (1) the careful feature design of weight statistics that captures the dominant information regarding weight distributions from pre-trained models, which potentially helps reduce noise from each task dataset; and (2) the joint training from all old

tasks on the task-specific teacher-distilled labels, enabling the implicit learning of task-agnostic and task-specific features that can benefit the generalization ability.

**Scaling Merging Tasks.** When the number of tasks was increased from 8 to 14 and eventually 20 (Wang et al., 2024b), *StatsMerging* continued to perform reliably, consistently surpassing prior merging approaches. This steady improvement highlights the method's ability to handle increasingly diverse task distributions. The trend persists across both ViT-B/32 and ViT-L/14 backbones, as illustrated in Table 3. Note RegMean++ (Huu-Tien et al., 2025) does not provide 14-task results.

Table 3: Comparison of different merging methods on the Vision Merging Benchmark (8, 14, and 20 tasks) with ViT-B/32 and ViT-L/14 backbones. Results of our method *StatsMerging* are shaded in gray. Bold and underscore indicate the highest and second-highest scores within the merging group below the double rules in each column, respectively. LW: Layer-wise. T: Task.

| Method | ViT-B/32 | | | ViT-L/14 | | |
|---|---|---|---|---|---|---|
| | 8T | 14T | 20T | 8T | 14T | 20T |
| Pre-Trained | 48.4 | 57.3 | 56.1 | 64.4 | 68.0 | 65.1 |
| Weight Averaging | 66.5 | 64.4 | 61.1 | 79.4 | 76.6 | 71.5 |
| Task Arithmetic | 70.8 | 65.4 | 60.6 | 84.8 | 79.3 | 74.0 |
| TIES-Merging | 75.1 | _68.0_ | 63.4 | 86.9 | _79.5_ | 75.7 |
| RegMean++ | _84.4_ | – | _77.0_ | _91.2_ | – | _81.0_ |
| LW *StatsMerging++* | **94.5** | **90.7** | **86.8** | **94.1** | **89.1** | **88.9** |

**Extension to Heterogeneous Architectures for Model Merging.** To the best of our knowledge, *StatsMerging* is the first to offer improved performance without the assumption of architectural identity as in prior works (Wortsman et al., 2022; Ilharco et al., 2023; Yadav et al., 2023a; Matena & Raffel, 2022; Jin et al., 2023). The procedure of Heterogeneous distillation is illustrated in Fig. 5. When a Task $k$ pre-trained model shared a different architecture (parallelogram) with the target architecture (rounded rectangle), we followed the steps in 3 (a) to generate pseudo labels to guide the training of the Task $k$ model with the target architecture (rounded rectangle in red). This enabled a direct integration into existing model merging pipeline as all models share the same target architecture after distillation. We conducted experiments on ResNet50 (RN) and ViT-B/32 (VT) to represent Convolutional Neural Network (CNN) and Vision Transformer (ViT) architectures.

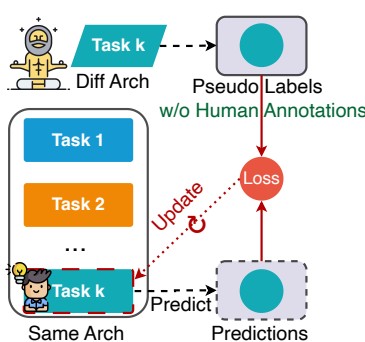

Figure 5: Heterogeneous distillation graph. Shapes represent architectures.

In particular, we distilled fine-tuned VT teachers into a RN (Khanuja et al., 2021) student on three diverse tasks of CIFAR-10 (CI), EuroSAT (EU), and Stanford Cars (CA) with the distillation loss:

$$\mathcal{L} = \alpha \mathcal{L}_{\text{CE}}(\hat{y}_k, \hat{y}) + (1 - \alpha) T^2 \mathcal{L}_{\text{KL}}\big(\sigma(\tfrac{z}{T}), \sigma(\tfrac{z_t}{T})\big), \quad (8)$$

where $\mathcal{L}_{\text{KL}}$ denotes KL-Divergence loss, $z$ is logit, $T = 4.0$ represents temperature, $\alpha = 0.7$ is the weight balance of two sub-losses. CI is used due to the available pre-trained RN weights. Remarkably, the distilled RN matches its VT teacher's accuracy, achieving $76.4\%$ (VT: $77.7\%$) for CA and $94.5\%$ for EU (VT: $99.7\%$) despite the architectural difference shown in Table 3. We then applied our *StatsMerging* to combine the CI–trained RN and its distilled variants. We merged multiple task models into a single RN using the merging coefficients inferred by *StatsMergeLearner*, yielding an $81.3\%$ Avg Acc, outperforming the vanilla Task-Arithmetic of $73.7\%$.

Table 4. Multi-task merging performance (Avg Acc %) of models in heterogeneous architectures: ResNet50 (RN) & ViT-B/32 (VT). *StatsMerging*: shaded in gray. MTL: Multitask Learning. MLD: Multitask Distilled.

| Method | CI | CA | EU | Avg Acc |
|---|---|---|---|---|
| Backbone | RN | VI | VI | - |
| Distilled | - | RN | RN | - |
| Individual | 97.8 | 77.7 | 99.7 | 91.7 |
| Distilled | - | 76.4 | 94.5 | - |
| MTL | 96.4 | 74.6 | 96.2 | 89.1 |
| MTD | 89.3 | 52.7 | 83.4 | 75.1 |
| Weight Averaging | 77.1 | 56.4 | 64.9 | 59.4 |
| TIES-Merging | 76.5 | 52.8 | 80.1 | 69.8 |
| Task Arithmetic | 81.4 | 61.6 | 78.2 | 73.7 |
| AdaMerging | 84.9 | 65.1 | 85.7 | 78.6 |
| WEMoE | 86.5 | 67.2 | 87.6 | 80.4 |
| **LW *StatsMerging*** | **87.2** | **68.4** | **88.4** | **81.3** |

### 4.3 *StatsMerging* ANALYSIS

**Statistical Feature Ablation Study.** We conduct an ablation study on the statistical features. Results in Table 4 show that combining all statistical features improves merging performance, validating our design choice. Notably, the singular values $\sigma'$ improve the multi-task performance in both same and different architecture settings by $+3.0$ and $+3.2$ increase of average accuracy, justifying our design choice of using SVD.

Table 5: Multi-task performance (Avg Acc %) of *StatsMerging* when ablating statistical features of ViT-B/32 (4) models on four tasks: CA, EU, RE & GT. Bold: top score. *StatsMerging*: shaded in gray.

| Same Architecture | | | | | | Different Architecture | | | | |
|---|---|---|---|---|---|---|---|---|---|---|
| $\mu$ | $\sigma^2$ | $m$ | $\sigma'$ | Avg Acc | | $\mu$ | $\sigma^2$ | $m$ | $\sigma'$ | Avg Acc |
| ✓ | | | | 83.4 | | ✓ | | | | 76.2 |
| ✓ | ✓ | | | 84.1 (+0.7) | | ✓ | ✓ | | | 77.5 (+1.3) |
| ✓ | ✓ | ✓ | | 87.2 (+3.1) | | ✓ | ✓ | ✓ | | 78.1 (+0.6) |
| ✓ | ✓ | ✓ | ✓ | **90.2 (+3.0)** | | ✓ | ✓ | ✓ | ✓ | **81.3 (+3.2)** |

**SVD Rank Study.** We analyze the impact of SVD rank on merging performance. Table 4 shows that using rank 3, which generally preserves more than 95% of the weight energy, yields the strongest overall results. This provides empirical support for our choice of rank.

Table 6: Impact of Rank on Multi-task merging performance (Avg Acc %) when merging StatsMerging++ ViT-B/32 models on eight vision tasks. Bold: top score. *StatsMerging*: shaded in gray.

| Rank | 1 | 2 | **3** | 4 | 5 |
|---|---|---|---|---|---|
| Avg Acc | 86.5 | 87.2 | **94.5** | 89.2 | 86.7 |

**Coefficient Analysis.** We visualize the heatmap of ViT-B/32 (4) across eight tasks in Fig. 6. We make several key observations: (1) the **common recurring pattern** of coefficients $\lambda$ across all eight tasks from earlier (left) to deeper (right) layers aligns with the repeated self-attention blocks in the ViT architecture, e.g. Multi-Head Self-Attention (MHSA), MLP (Feed-Forward Network), and LayerNorm, etc, demonstrating the need of various coefficients for various types of layers; (2) The **sparse non-uniform coefficient distributions** (various colors like Layer 13, 19 or 25) suggests that merging layers can be more efficient at some specific layers instead of using one coefficient for an entire pre-trained model, justifying the our granularity choice of Layer-Wise over Task-Wise level; (3) some **task-specific coefficient distributions** verify the necessity of assigning distinct merging coefficients across tasks in various layers, such as in Layer 5 vs. 147. Such distributions reflect the various visual representations for different semantics learned across both layers and tasks.

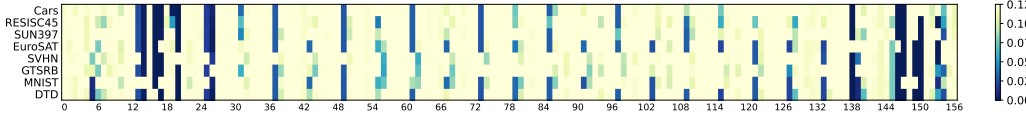

Figure 6: Heatmap of *StatsMerging* merging coefficients $\lambda$ of ViT-B/32 (4) across eight tasks. X-axis: layer index. Y-axis: Tasks. Coefficients are normalized to sum to 1.

## 5 CONCLUSION

We propose *StatsMerging*, a novel merging technique without human annotations. The key intuition lies in the guidance of weight statistics using a lightweight MLP learner, *StatsMergeLearner*, to learn merging coefficient prediction. Exhaustive experiments demonstrate the effectiveness of our proposed *StatsMerging* in model merging in diverse Vision and NLP tasks.

## 6 ETHICS STATEMENT

This work focuses on a method for merging pre-trained models using statistical guidance, with the goal of reducing memory redundancy and improving efficiency in multi-task deployments. Our research does not involve human subjects, personally identifiable information, or sensitive data. All experiments are conducted on publicly available benchmark datasets, following their intended academic usage licenses.

We recognize that model merging and multi-task deployment systems could potentially be misused to scale applications without considering fairness, robustness, or downstream societal impacts. To mitigate these risks, we limit our evaluation to standard academic benchmarks and encourage practitioners to carefully assess bias, fairness, and safety when applying such methods in real-world settings.

## 7 REPRODUCIBILITY STATEMENT

We make every effort to ensure the reproducibility of our results. All experiments were run on publicly available datasets (e.g., RESISC45, EuroSAT, CIFAR-100, etc.), and we describe dataset preprocessing, training, and evaluation protocols in detail in the main paper and appendix. Hyperparameters, model architectures, and training schedules are fully specified.

Our method requires only pre-trained models. No additional training data beyond the standard benchmarks is used. To facilitate replication, We attached training and test code github link for reproducing results. We included all details of GPU Hyperpatameters used in experiments.

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

# A APPENDIX

## A.1 EXPERIMENT SETTINGS

This section presents a comprehensive overview of the datasets, baseline methods, and training procedures.

**Task.** A task is referred to the specific problem or objective that a model is designed to solve. In this paper, a task is defined as classifying images within a given dataset.

**Dataset Details.** This study follows the multi-task model merging protocol from Task Arithmetic (Ilharco et al., 2023), TIES-Merging (Yadav et al., 2023a) and AdaMerging (Yang et al., 2023) on eight image classification datasets. The details are provided below:

**Vision Datasets:**

- **SUN397 (SU)** (Xiao et al., 2016): a scene classification dataset consisting of 397 classes and a total of 108,754 images, with each class containing a minimum of 100 images.
- **Stanford Cars (CA)** (Krause et al., 2013): a car classification benchmark dataset comprosing 196 categories and 16,185 images in total. For each category, the dataset is evenly divided into training and test sets in a 1:1 ratio.
- **RESISC45 (RE)** (Cheng et al., 2017): a remote sensing image scene classification benchmark with 45 scene classes and 31,500 images. Approximately 700 images are included in each class.
- **EuroSAT (EU)** (Helber et al., 2019): a 10-class satellite image classification dataset with 27,000 labeled and geo-referenced images.
- **SVHN (SV)** (Netzer et al., 2011): a real-world digit classification dataset derived from house numbers in Google Street View images. This datasets consists of 10 classes with 73,257 training samples and 26,032 test samples. Additional 531,131 samples are available for training.
- **GTSRB (GT)** (Stallkamp et al., 2011): a traffic sign classification dataset consisting of 43 classes and more than 50,000 samples in total.
- **MNIST (MN)** (LeCun et al., 1998): a benchmark dataset for image classification, containing grayscale images of handwritten digits across 10 classes. It includes 60,000 training and 10,000 test images, with a balanced number across classes.
- **DTD (DT)** (Cimpoi et al., 2014): a texture classification dataset consisting of 47 classes and a total of 5,640 images, with approximately 120 images per class.

**NLP Datasets:**

- **PAWS (PA) – Paraphrase Adversaries from Word Scrambling** (Zhang et al., 2019): a challenging paraphrase identification dataset with over 108,463 sentence pairs. It contains adversarially-generated non-paraphrases with high lexical overlap to test a model's semantic understanding beyond simple word-matching heuristics.
- **QASC (QA) – Question Answering via Sentence Composition** (Khot et al., 2020): a multi-hop question-answering dataset with nearly 10,000 multiple-choice science questions. It is designed to test compositional reasoning, requiring models to combine two distinct facts to find the answer, often by reasoning over intermediate concepts not mentioned in the question.
- **Quartz (QU)** (Tafjord et al., 2019): a dataset of nearly 4,000 questions focused on qualitative reasoning from text. Each question requires reasoning about the relationship between two concepts and is presented with two candidate answers. The dataset is designed to test a deeper understanding that goes beyond simple fact retrieval.
- **Story Cloze (SC)** (Mostafazadeh et al., 2016): a commonsense reasoning dataset for evaluating story comprehension. Which contains 50,000 five-sentence stories about everyday life. The dataset involves reading a four-sentence story context and choosing the correct, causally sound ending from two possible options. This requires a model to understand narrative flow and commonsense implications.

- **WikiQA (WQ)** (Yang et al., 2015): an open-domain question-answering dataset for the task of answer sentence selection, featuring over 3,000 . For each question, which is sourced from Bing query logs, a set of candidate sentences are extracted from Wikipedia. The goal is to identify which of the sentences actually contains the answer to the question.

- **Winogrande (WG)** (Sakaguchi et al., 2020): a large-scale commonsense reasoning dataset of 44,000 problems, inspired by the Winograd Schema Challenge. The task is pronoun resolution, where a model must resolve an ambiguous pronoun in a sentence. The dataset was constructed using an adversarial filtering process to remove biases and create problems that are more difficult for statistical models.

- **WSC (WS) – Winograd Schema Challenge** (Levesque et al., 2012): a benchmark dataset for commonsense reasoning focused on pronoun resolution, total 273 problems. It consists of pairs of sentences that differ by only a few words, which completely changes the referent of an ambiguous pronoun. Correctly resolving the pronoun requires world knowledge and reasoning capabilities.

**Baseline Details.** We evaluate performance using eight comparison baselines and four alternative configurations of our method.

- **Individual**: Each task is handled by an independently fine-tuned model with no interference between tasks. However, this approach cannot perform multiple tasks simultaneously.

- **Traditional MTL**: This approach aggregates the original training data from all tasks to train a single multi-task model. It serves as a reference *upper bound* for evaluating model merging performance.

- **Weight Averaging**: A simple model merging technique that averages the parameters of multiple models directly. It is typically considered a *lower bound* for model merging performance.

- **Fisher Merging** (Matena & Raffel, 2022): This method computes the Fisher Information Matrix to assess parameter importance, guiding the model merging process based on these importance scores.

- **RegMean** (Jin et al., 2023): Introduces a regularization constraint during merging, enforcing the $L_2$ distance between the merged model and individual models to remain small.

- **Task Arithmetic** (Ilharco et al., 2023): This method is the first to propose the concept of "task vectors" and merges these vectors into a pre-trained for model merging.

- **TIES-Merging** (Yadav et al., 2023a): Addresses task conflict in Task Arithmetic (Ilharco et al., 2023) by removing redundant parameters and resolving sign conflicts through a three-step procedure: Trim, Elect Sign, and Disjoint Merge.

- **EMR-MERGING** (Huang et al., 2024): This approach is a tuning-free method that merges models in three steps, by selecting a unified parameter sign (Elect), aligning task-specific parameters via masking (Mask), and adjusting their magnitudes with task-specific scaling factors (Rescale).

- **AdaMerging** (Yang et al., 2023): Builds on Task Arithmetic (Ilharco et al., 2023) by employing an unsupervised method to automatically learn merging coefficients for each task vector.

- **AdaMerging++** (Yang et al., 2023): An extension of TIES-Merging (Yadav et al., 2023a) that uses an unsupervised approach to learn task-specific merging coefficients.

- *StatsMerging* (**Ours**): A lightweight learning-based method guided by the weight distribution statistical features (stats) of task-specific pre-trained weight models, including the mean, variance, magnitude and singular values. This method employs *StatsMergeLearner* o learn stats by knowledge distillation from task-specific teachers without manual labels.

- *StatsMerging++* (**Ours**): A more extensively trained version of *StatsMerging*.

**Training Details.**

- **Task-Specific Teacher**: For each task, we utilize its corresponding **Individual** model as the **Teacher**.

Code is available at https://github.com/statsmerging/statsmerging.

## A.2 DETAILS OF TASK-LEVEL RESULTS

We present the details of task-level results in this section, demonstrating ViT-B/32, ViT-L/14 for Vision tasks and T5 Base, T5 Large for NLP tasks in Fig. 7 and Tables 3, 4, 5, and 6.

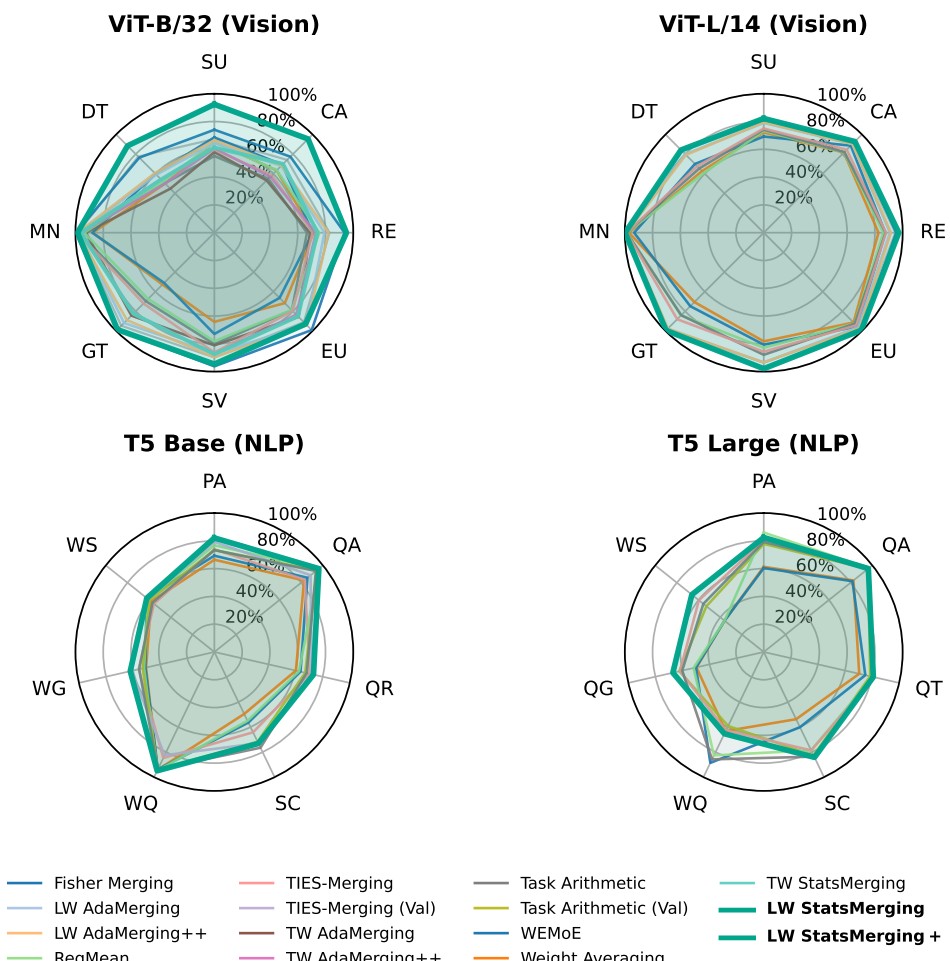

Figure 7: *StatsMerging* achieved state-of-the-art performance across scales (ViT-B/32, ViT-L/14, T5 Base, and T5 Large) in both Vision (top) and NLP (bottom) benchmarks.

## A.2.1 VISION BENCHMARK

Table 3: Multi-task merging performance (Avg Acc %) when merging **ViT-B/32** models on eight **vision** tasks. Results of our method *StatsMerging* are shaded in gray. Bold and underscore indicate the highest and second-highest scores within the merging group below the double rules in each column, respectively. TW: Task-wise. LW: Layer-wise.

| Method | SU | CA | RE | EU | SV | GT | MN | DT | Avg Acc |
|---|---|---|---|---|---|---|---|---|---|
| Pre-Trained | 62.3 | 59.7 | 60.7 | 45.5 | 31.4 | 32.6 | 48.5 | 43.8 | 48.0 |
| Individual | 75.3 | 77.7 | 96.1 | 99.7 | 97.5 | 98.7 | 99.7 | 79.4 | 90.5 |
| Traditional MTL | 73.9 | 74.4 | 93.9 | 98.2 | 95.8 | 98.9 | 99.5 | 77.9 | 88.9 |
| Weight Averaging | 65.3 | 63.4 | 71.4 | 71.7 | 64.2 | 52.8 | 87.5 | 50.1 | 65.8 |
| Task Arithmetic | 55.2 | 54.9 | 66.7 | 78.9 | 80.2 | 69.7 | 97.3 | 50.4 | 69.1 |
| Fisher Merging | 68.6 | 69.2 | 70.7 | 66.4 | 72.9 | 51.1 | 87.9 | 59.9 | 68.3 |
| RegMean | 65.3 | 63.5 | 75.6 | 78.6 | 78.1 | 67.4 | 93.7 | 52.0 | 71.8 |
| TIES-Merging | 59.8 | 58.6 | 70.7 | 79.7 | 86.2 | 72.1 | 98.3 | 54.2 | 72.4 |
| TW AdaMerging | 58.0 | 53.2 | 68.8 | 85.7 | 81.1 | 84.4 | 92.4 | 44.8 | 71.1 |
| TW AdaMerging++ | 60.8 | 56.9 | 73.1 | 83.4 | 87.3 | 82.4 | 95.7 | 50.1 | 73.7 |
| **TW *StatsMerging*** | 61.3 | 70.0 | 74.2 | 85.2 | 87.5 | 82.5 | 96.2 | 54.2 | 76.4 (+3.3) |
| LW AdaMerging | 64.5 | 68.1 | 79.2 | 93.8 | 87.0 | 91.9 | 97.5 | 59.1 | 80.1 |
| LW AdaMerging++ | 66.6 | 68.3 | 82.2 | 94.2 | 89.6 | 89.0 | 98.3 | 60.6 | 81.1 |
| WEMoE | 74.1 | 77.4 | 93.7 | **99.1** | **96.2** | **98.9** | **99.6** | 76.4 | 89.4 |
| **LW *StatsMerging*** | 67.4 | 74.1 | 82.9 | 91.1 | 89.8 | 94.7 | 98.3 | 77.5 | 84.5 |
| **LW *StatsMerging*++** | **92.4** | **95.4** | **95.1** | 92.9 | 94.6 | 98.7 | 98.5 | **88.4** | **94.5 (+5.1)** |

Table 4: Multi-task merging performance (Avg Acc %) when merging **ViT-L/14** models on eight **vision** tasks. Results of our method *StatsMerging* are shaded in gray. Bold and underscore indicate the highest and second-highest scores within the merging group below the double rules in each column, respectively. TW: Task-wise. LW: Layer-wise.

| Method | SU | CA | RE | EU | SV | GT | MN | DT | Avg Acc |
|---|---|---|---|---|---|---|---|---|---|
| Pre-Trained | 68.2 | 77.9 | 71.3 | 61.3 | 58.4 | 50.6 | 76.4 | 55.4 | 64.9 |
| Individual | 82.3 | 92.4 | 97.4 | 99.9 | 98.1 | 99.2 | 99.7 | 84.1 | 94.1 |
| Traditional MTL | 80.8 | 90.6 | 96.3 | 96.3 | 97.6 | 99.1 | 99.6 | 84.4 | 93.5 |
| Weight Averaging | 72.1 | 81.6 | 82.6 | 91.4 | 78.2 | 70.6 | 97.0 | 62.8 | 79.5 |
| Fisher Merging | 69.2 | 88.6 | 87.5 | 95.5 | 80.6 | 74.8 | 93.3 | 70.0 | 82.2 |
| RegMean | 73.3 | 81.8 | 86.1 | 92.4 | 82.8 | 84.2 | 98.5 | 60.8 | 82.5 |
| Task Arithmetic | 74.1 | 82.1 | 87.7 | 92.6 | 87.9 | 84.0 | 98.6 | 65.5 | 84.4 |
| TIES-Merging | 75.0 | 84.5 | 88.0 | 94.3 | 85.7 | 88.1 | 98.7 | 67.7 | 84.5 |
| LW AdaMerging | 79.0 | 90.3 | 90.8 | 96.2 | 93.4 | 98.0 | 99.0 | 79.9 | 90.8 |
| LW AdaMerging++ | 79.4 | 90.3 | 91.6 | 97.4 | 93.4 | 97.6 | 99.0 | 79.2 | 91.0 |
| WEMoE | 81.4 | 92.6 | 95.4 | **99.4** | 97.7 | **99.9** | 99.7 | 83.7 | 93.6 |
| **LW *StatsMerging*** | 80.6 | 90.5 | 94.7 | 96.8 | 93.6 | 98.3 | 98.9 | 83.2 | 92.1 |
| **LW *StatsMerging*++** | **82.2** | **92.8** | **97.2** | 99.3 | **97.9** | 99.5 | **99.8** | **84.2** | **94.1 (+0.5)** |

## A.2.2 NLP BENCHMARK

Table 5: Evaluation of model merging methods on seven **NLP** tasks on **T5 Base** Models. Results of our method *StatsMerging* are shaded in gray. Bold and underline indicate the highest and second-highest scores within the merging group below the double rules in each column, respectively.

| Method | Val | PA | QA | QR | SC | WQ | WG | WS | Avg Acc |
|---|---|---|---|---|---|---|---|---|---|
| Pre-Trained | – | 49.9 | 35.8 | 53.3 | 48.1 | 76.2 | 50.0 | 61.1 | 53.5 |
| Individual | – | 94.3 | 98.3 | 80.4 | 84.7 | 95.5 | 64.1 | 62.5 | 82.8 |
| Traditional MTL | – | 94.0 | 97.9 | 82.5 | 86.7 | 95.0 | 64.1 | 65.3 | 83.6 |
| Weight Averaging | ✗ | 66.4 | 82.6 | 60.2 | 49.5 | 94.1 | 50.4 | 58.3 | 65.9 |
| Task Arithmetic | ✗ | 73.3 | 93.5 | 68.2 | **76.5** | 93.7 | 55.5 | 56.9 | 73.9 |
| TIES-Merging | ✗ | 74.0 | 83.3 | 70.3 | 64.2 | 84.7 | 55.9 | 55.6 | 69.7 |
| Fisher Merging | ✓ | 69.3 | 85.7 | 63.6 | 56.4 | 93.8 | 50.9 | **62.5** | 68.9 |
| RegMean | ✓ | 76.8 | **96.2** | 62.5 | 55.0 | 94.8 | 51.9 | 61.1 | 71.2 |
| Task Arithmetic | ✓ | 73.4 | 94.3 | 67.1 | 71.7 | 94.1 | 52.9 | 59.7 | 73.2 |
| TIES-Merging | ✓ | 79.3 | 88.6 | 71.8 | 72.9 | 82.5 | 61.3 | 61.1 | 73.9 |
| LW *StatsMerging* | ✓ | **82.1** | 96.2 | **73.2** | 73.1 | **94.9** | 62.1 | 62.2 | **77.6 (+3.7)** |

Table 6: Evaluation of merging methods across seven **NLP** tasks on **T5 Large** Models. Results of our method *StatsMerging* are shaded in gray. Bold and underline indicate the highest and second-highest scores within the merging group below the double rules in each column, respectively.

| Method | Val | PA | QA | QT | SC | WQ | QG | WS | Avg Acc |
|---|---|---|---|---|---|---|---|---|---|
| Pre-Trained | – | 55.4 | 14.3 | 54.1 | 54.1 | 71.0 | 49.3 | 63.9 | 51.7 |
| Individual | – | 94.4 | 98.9 | 87.8 | 90.8 | 96.0 | 74.7 | 79.2 | 88.8 |
| Traditional MTL | – | 94.2 | 98.5 | 89.3 | 92.0 | 95.4 | 73.5 | 73.6 | 88.1 |
| Weight Averaging | ✗ | 61.3 | 82.6 | 70.5 | 53.7 | 63.2 | 49.7 | 36.1 | 59.6 |
| Task Arithmetic | ✗ | 79.2 | 96.8 | 80.2 | 83.6 | 85.8 | 60.2 | 55.6 | 73.5 |
| TIES-Merging | ✗ | 80.5 | 96.2 | 81.8 | 78.6 | 62.6 | 61.9 | 59.7 | 74.4 |
| Fisher Merging | ✓ | 60.4 | 81.7 | 75.0 | 60.1 | **88.6** | 50.0 | 36.1 | 64.6 |
| RegMean | ✓ | **86.0** | **96.9** | 80.7 | 78.6 | 82.6 | 51.8 | 36.1 | 73.2 |
| Task Arithmetic | ✓ | 77.8 | 96.0 | 78.6 | 82.6 | 59.1 | 62.3 | 52.8 | 73.3 |
| TIES-Merging | ✓ | 81.5 | 96.2 | 80.1 | 83.6 | 64.9 | 66.5 | 65.3 | 76.9 |
| LW *StatsMerging* | ✓ | 82.4 | 96.3 | **80.9** | **84.2** | 65.3 | **67.1** | 66.2 | **77.5 (+0.6)** |

### A.3 DETAILS OF TASK-SPECIFIC TEACHER DISTILLATION

1. **Task-Specific Teacher Models Preparation.** Collect $K$ pre-trained models $\Theta = \{\theta_1, \theta_2, \ldots, \theta_k\}$, where each model weight is fine-tuned on an independent task $k$ with dataset $\{x_i, y_i\}_k \in D_k$. $D_k$ denotes the dataset for task $k$, $x_i$ and $y_i$ represent a sample's input and its corresponding label. Note that $y_i$ is not used for SML learning but only in the evaluation step.

2. **Train/Val/Test Split.** Each dataset $D_k$ for task $k$ is split into training, validation, and test sets with an 8:1:1 ratio unless otherwise specified, denoted as $D_k^{\text{train}}$, $D_k^{\text{val}}$, and $D_k^{\text{test}}$, respectively.

3. **Pseudo Label Preparation for Training Set $D^{\text{train}}$.** Following (2), for task $k$, the task-specific teacher $\theta_k$ takes a sample $x_{i,k}$ and generates its prediction $\hat{y}_{i,k}$ as a pseudo label. The resulting pairs $(x_{i,k}, \hat{y}_{i,k})$ are aggregated to form task $k$'s training dataset $D_k^{\text{train}} \subseteq D^{\text{train}}$.

4. **Val $D^{\text{val}}$ and Test $D^{\text{test}}$ Preparation.** Following (2), for task $k$, the original pairs $(x_{i,k}, y_{i,k})$ in the split validation set ($D_k^{\text{val}} \subseteq D^{\text{val}}$) or test set ($D_k^{\text{test}} \subseteq D^{\text{test}}$) are used, where $y_{i,k}$ is the human-annotated ground truth label used solely for evaluation.

5. **Complete Dataset $D$ Preparation.** Aggregate $D^{\text{train}}$, $D^{\text{val}}$, and $D^{\text{test}}$ to form the complete dataset $D = \{D^{\text{train}}, D^{\text{val}}, D^{\text{test}}\}$.

Concretely, in the eight vision tasks, the samples $\{x_i, y_i\}_k \in D_k$ are drawn from the following datasets: SUN397 (SU), Cars (CA), RESISC45 (RE), EuroSAT (EU), SVHN (SV), GTSRB (GT), MNIST (MN), and DTD (DT). The pseudo label $\hat{y}_i$ is generated by the task-specific teacher set $\Theta = \{\theta_1, \theta_2, \ldots, \theta_k\}$. These are aggregated to constitute the overall dataset $D$. The same procedure applies to the NLP tasks.

### A.3.1 Details of SVD Construction

The construction of the parameter matrix for singular value decomposition (SVD) is as follows: For each layer $k$, we flatten its parameter tensor Wk into a 2D matrix. For linear layers, this is typically (out features, in features). For convolutional layers with a kernel of shape (out channels, in channels, kernel height, kernel width), we reshape it to (out channels, in channels × kernel height × kernel width). We then compute the SVD:

$$W_K = U_K \Sigma_K V_K^T \tag{9}$$

and extract the singular values from $\Sigma_K$. The singular values across all layers are concatenated to form the feature vector used as input to SML.

## A.4 Theoretical Analyses

### A.4.1 Optimization Perspective

Following the setup in Sec.3.2, let $\{\theta_k\}_{k=1}^K$ be $K$ pre-trained models and

$$\theta(\lambda) = \sum_{k=1}^K \lambda_k \theta_k, \qquad \lambda \in \Delta^{K-1}, \tag{10}$$

where $\Delta^{K-1}$ is the $(K-1)$-dimensional probability simplex.

Since ground-truth labels are unavailable, we train using teacher pseudo labels $q(y \mid x)$. Following (Bishop, 2006), replacing the true label distribution by any surrogate distribution yields a valid expected log-likelihood objective. Thus the pseudo label cross-entropy is

$$\mathcal{L}_{\text{PL}}(\theta) = \mathbb{E}_{x \sim \mathcal{D}} \mathbb{E}_{y \sim q(\cdot|x)} \big[ -\log p_\theta(y \mid x) \big]. \tag{11}$$

Using the standard derivative of log-likelihood,

$$\nabla_\theta \mathcal{L}_{\text{PL}}(\theta) = \mathbb{E}_{x,y \sim q} \left[ -\nabla_\theta \log p_\theta(y \mid x) \right]. \tag{12}$$

Motivated by the classical Fisher Information (Bishop, 2006), we define the *pseudo label (PL) Fisher*:

$$F_{\text{PL}}(\theta) = \mathbb{E}_{x,y \sim q} \left[ \nabla_\theta \log p_\theta(y \mid x) \nabla_\theta \log p_\theta(y \mid x)^\top \right]. \tag{13}$$

When $q(\cdot \mid x) = p^\star(\cdot \mid x)$, this reduces to the standard Fisher Information matrix.

For a reference model $\theta_0$, the second-order Taylor expansion (Boyd & Vandenberghe, 2004) yields:

$$\mathcal{L}_{\text{PL}}(\theta) \approx \mathcal{L}_{\text{PL}}(\theta_0) + \frac{1}{2}(\theta - \theta_0)^\top H_{\text{PL}}(\theta_0)(\theta - \theta_0). \tag{14}$$

For cross-entropy networks near optimum, the Hessian is well approximated by the Fisher (Bishop, 2006; Martens, 2014):

$$H_{\text{PL}}(\theta_0) \approx F_{\text{PL}}(\theta_0). \tag{15}$$

Thus:

$$\mathcal{L}_{\text{PL}}(\theta) \approx \mathcal{L}_{\text{PL}}(\theta_0) + \frac{1}{2}(\theta - \theta_0)^\top F_{\text{PL}}(\theta_0)(\theta - \theta_0). \tag{16}$$

Define the parameter-difference matrix:

$$P = \big[\theta_1 - \theta_0, \; \theta_2 - \theta_0, \; \ldots, \; \theta_K - \theta_0\big] \in \mathbb{R}^{n \times K} \tag{17}$$

where $n$ is the number of parameters in $\theta$.

Since
$$\theta(\lambda) - \theta_0 = P\lambda, \tag{18}$$
substituting into equation 16 gives:
$$\mathcal{L}_{\mathrm{PL}}(\theta(\lambda)) \approx \mathcal{L}_{\mathrm{PL}}(\theta_0) + \frac{1}{2}\lambda^\top \underbrace{(P^\top F_{\mathrm{PL}}(\theta_0)P)}_{A_{\mathrm{PL}}}\lambda. \tag{19}$$

Thus the optimal merging coefficients are:
$$\lambda^\star_{\mathrm{PL}} = \arg\min_{\lambda \in \Delta^{K-1}} \frac{1}{2}\lambda^\top A_{\mathrm{PL}}\lambda. \tag{20}$$

Let $p^\star(y \mid x)$ denote the true label distribution. The true Fisher and true quadratic matrix is defined as:
$$F_{\mathrm{true}}(\theta_0) = \mathbb{E}_{x,y\sim p^\star}[\nabla \log p_\theta \nabla \log p_\theta^\top], \tag{21}$$

$$A_{\mathrm{true}} = P^\top F_{\mathrm{true}}(\theta_0)P. \tag{22}$$

Assume the teacher satisfies for total variation:
$$\mathrm{TV}(q(\cdot \mid x), p^\star(\cdot \mid x)) \le \varepsilon, \tag{23}$$
and that likelihood gradients are bounded (Shalev-Shwartz & Ben-David, 2014; van der Vaart, 1998).

Standard stability arguments yield:
$$\|F_{\mathrm{PL}}(\theta_0) - F_{\mathrm{true}}(\theta_0)\| = O(\varepsilon). \tag{24}$$
Thus:
$$\|A_{\mathrm{PL}} - A_{\mathrm{true}}\| = \|P^\top(F_{\mathrm{PL}} - F_{\mathrm{true}})P\| = O(\varepsilon). \tag{25}$$

From sensitivity analysis of strictly convex quadratic programs (Boyd & Vandenberghe, 2004):
$$\|\lambda^\star_{\mathrm{PL}} - \lambda^\star_{\mathrm{true}}\| = O(\varepsilon). \tag{26}$$

Taking all together, the above derivation shows that pseudo label supervision is theoretically sufficient for recovering the Fisher-optimal merging coefficients. When the teacher pseudo label distribution is close to the ground truth label distribution in total variation distance, the pseudo label Fisher curvature approximates the true Fisher curvature, and the resulting quadratic program yields merging weights provably within $O(\epsilon)$ of the ground truth solution. Thus. SML trained with pseudo labels optimizes nearly the same second-order objective if ground truth labels were available.

Furthermore, our statistics of mean, variance, magnitude, and rank 3 from SVD serve as compact, data-free approximations to the Fisher curvature that governs the optimal merge. The variance term captures diagonal Fisher structure, the mean and magnitude encode parameter scale and shift effects that influence the quadratic form $P^T F P$, and the low-rank SVD directions approximate dominant Fisher eigenvectors observed empirically in deep networks. Thus, the statistic vector $S_k$ preserves the key curvature signals needed for SML to learn merging coefficients that closely match the Fisher-optimal solution.

### A.4.2 MEAN AND VARIANCE DETERMINE THE SECOND-MOMENT

Define mean and variance of a fine-tuned weight $\theta$.
$$\mu = \mathbb{E}[\theta] \tag{27}$$

$$\sigma^2 = \mathrm{Var}(\theta) = \mathbb{E}[(\theta - \mu)^2]. \tag{28}$$
Then the second moment of $\theta$ can be written as

$$\mathbb{E}[(\theta)^2] = \mathbb{E}\big[(\theta - \mu + \mu)^2\big]$$

$$= \mathbb{E}\big[(\theta - \mu)^2\big] + 2\mu\mathbb{E}[\theta - \mu] + \mu^2$$

$$= \underbrace{\mathbb{E}\big[(\theta - \mu)^2\big]}_{\sigma^2} + 2\mu \cdot 0 + \mu^2 \tag{29}$$

$$= \sigma^2 + \mu^2.$$

For every parameter $\theta$,

$$\mathbb{E}[\theta^2] \;=\; \mathrm{Var}(\theta) \;+\; (\mathbb{E}[\theta])^2 \;=\; \sigma^2 + \mu^2. \tag{30}$$

Thus, the mean and variance of a parameter fully determine its second moment (Papoulis, 1965):

$$\text{second moment of } \theta = \mathbb{E}[\theta^2], \tag{31}$$

and therefore offer a complete, data-free representation of the second-order statistics underlying the parameter distribution.

## A.5 EXTENDED EXPERIMENTS

### A.5.1 MERGING PERFORMANCE

Extended experimental merging results are presented in Table 7. Results for Pre-Trained models, Individual models, and those trained using Traditional MTL are listed above the double horizontal lines. Below these lines, the comparison is organized into three groups: Task-wise methods appear first, followed by Layer-wise approaches, and finally the Parameter-wise method. Notably, while finer granularity is generally associated with improved merging performance (Yang et al., 2023), our **LW *StatsMerging*++**, operating at the Layer-wise level, surpasses EMR-Merging (Huang et al., 2024), which is based on the finer Parameter-wise granularity.

Table 7: Multi-task merging performance (Avg Acc %) when merging ViT-B/32 models on eight tasks. Results of our method *StatsMerging* are shaded in gray. Bold and underscore indicate the highest and second-highest scores within the merging group below the double rules in each column, respectively. GL: Granularity Level. TW: Task-wise. LW: Layer-wise. PW: Parameter-wise.

| Method | SU | CA | RE | EU | SV | GT | MN | DT | Avg Acc |
|---|---|---|---|---|---|---|---|---|---|
| Pre-Trained | 62.3 | 59.7 | 60.7 | 45.5 | 31.4 | 32.6 | 48.5 | 43.8 | 48.0 |
| Individual | 75.3 | 77.7 | 96.1 | 99.7 | 97.5 | 98.7 | 99.7 | 79.4 | 90.5 |
| Traditional MTL | 73.9 | 74.4 | 93.9 | 98.2 | 95.8 | 98.9 | 99.5 | 77.9 | 88.9 |
| **Task-wise** | | | | | | | | | |
| Weight Averaging | 65.3 | 63.4 | 71.4 | 71.7 | 64.2 | 52.8 | 87.5 | 50.1 | 65.8 |
| Task Arithmetic | 55.2 | 54.9 | 66.7 | 78.9 | 80.2 | 69.7 | 97.3 | 50.4 | 69.1 |
| Fisher Merging | 68.6 | 69.2 | 70.7 | 66.4 | 72.9 | 51.1 | 87.9 | 59.9 | 68.3 |
| RegMean | 65.3 | 63.5 | 75.6 | 78.6 | 78.1 | 67.4 | 93.7 | 52.0 | 71.8 |
| TIES-Merging | 59.8 | 58.6 | 70.7 | 79.7 | 86.2 | 72.1 | 98.3 | 54.2 | 72.4 |
| TW AdaMerging | 58.0 | 53.2 | 68.8 | 85.7 | 81.1 | 84.4 | 92.4 | 44.8 | 71.1 |
| TW AdaMerging++ | 60.8 | 56.9 | 73.1 | 83.4 | 87.3 | 82.4 | 95.7 | 50.1 | 73.7 |
| **TW *StatsMerging*** | 61.3 | 70.0 | 74.2 | 85.2 | 87.5 | 82.5 | 96.2 | 54.2 | 76.4 |
| **Layer-wise** | | | | | | | | | |
| LW AdaMerging | 64.5 | 68.1 | 79.2 | 93.8 | 87.0 | 91.9 | 97.5 | 59.1 | 80.1 |
| LW AdaMerging++ | 66.6 | 68.3 | 82.2 | **94.2** | 89.6 | 89.0 | 98.3 | 60.6 | 81.1 |
| **LW *StatsMerging*** | 67.4 | 74.1 | 82.9 | 91.1 | 89.8 | 94.7 | 98.3 | 77.5 | 84.5 |
| LW *StatsMerging*++ | **92.4** | **95.4** | **95.1** | 92.9 | **94.6** | **98.7** | **98.5** | **88.4** | **94.5 (+13.4)** |
| **Parameter-wise** | | | | | | | | | |
| EMR-MERGING | 75.2 | 72.8 | 93.5 | 99.5 | 96.9 | 98.1 | 99.6 | 74.4 | 88.7 |

### A.5.2 GENERALIZATION EVALUATION OF SML

We use SML trained on the eight vision datasets (LW *StatsMerging*), where it was exposed solely to the ViT-B/32 architecture for Task-Specific Experts on each vision task. This SML is then used to generate merging coefficients for merging two *unseen* ResNet50 models, each pre-trained on *unseen* CIFAR10 (CF10) and CIFAR100 (CF100) tasks. We evaluate SML in the Layer-Wise (LW) setting. This setup is summarized in Table 8.

Table 8: Generalization Experiment Setup

|  | **Architecture Type** | **Architecture** | **Task** |
|---|---|---|---|
| Train | Task-Expert | ViT-B/32 | SU, CA, RE, EU, SV, GT, MN, DT |
| Test | Merged | ResNet50 | CF10, CF100 |

*Challenge: Mismatch Layer.* To generalize to a different architecture, we encountered the challenge that the expert and the merged model layers differ. We subsample 22 coefficients to merge ResNet50 models from the 320 ViT-B/32 coefficients, enforcing consistency in the relative positions of coefficients and layers across both architectures. This approach is inspired by the insight from LiNeS (Wang et al., 2024a) that common and task-specific features are learned in shallow and deeper layers, respectively. Preserving these relative positions may help maintain the common-to-task-specific relationship.

Results are shown in Table 9. To the best of our knowledge, we are the **first** to evaluate generalizability to an *unseen architecture*, as prior model merging methods assume identical model architectures. The pre-trained models achieved an Avg Acc of 85.97%. However, there remains a substantial gap between the pre-trained models (85.97%) and recent advanced merging methods, with LW AdaMerging and LW StatsMerging achieving 26.66% and 43.15%, respectively. This gap highlights the extremely challenging nature of the task, as both the test tasks and the merged model architecture are unseen. Notably, our proposed LW StatsMerging improves LW AdaMerging by a large margin of 16.49%.

Table 9: Multi-task merging performance (Avg Acc %) when merging ResNet50 models on CIFAR10 and CIFAR100 using SML trained with ViT-B/32 architecture on eight tasks. Results of our method STATSMERGING are in bold shaded in gray. **LW:** Layer-wise.

| **Method** | **CF10** | **CF100** | **Avg Acc** |
|---|---|---|---|
| Pre-Trained | 97.80 | 74.14 | 85.97 |
| LW AdaMerging | 44.21 | 9.10 | 26.66 |
| **LW *StatsMerging*** | **64.70 (+20.49)** | **21.60 (+12.50)** | **43.15 (+16.49)** |

### A.5.3 ROBUSTNESS EVALUATION

**Input Corruption Tolerance.** We evaluate the robustness of *StatsMerging* against Task Arithmetic (Ilharco et al., 2023) and AdaMerging (Yang et al., 2023) under three image corruption scenarios: Motion Blur, Impulse Noise, and Gaussian Noise. The corrupted test sets are constructed following the protocols outlined in (Yang et al., 2023; Hendrycks & Dietterich, 2019). We assess performance on four datasets: Stanford Cars (CA) (Krause et al., 2013), EuroSAT (EU) (Helber et al., 2019), RESISC45 (RE) (Cheng et al., 2017), and GTSRB (GT) (Stallkamp et al., 2011). Results are reported in Table 10. Overall, *StatsMerging* consistently outperforms the baselines. On the clean test set, it achieves a 2.4% accuracy improvement over AdaMerging. Under corrupted conditions, *StatsMerging* yields performance gains of 3.1%, 6.3%, and 4.3% for Motion Blur, Impulse Noise, and Gaussian Noise, respectively.

Table 10: Robustness results when merging ViT-B/32 models on four tasks. *StatsMerging*: shaded in gray. Bold: top score. Values are reported in %.

| Method | CA | EU | RE | GT | Avg Acc |
|---|---|---|---|---|---|
| **Clean Test Set** | | | | | |
| Task Arithmetic | 66.9 | 94.7 | 82.6 | 75.1 | 79.8 |
| AdaMerging | 73.7 | 96.1 | 85.8 | 96.3 | 88.0 |
| *StatsMerging* | **75.6** | **96.3** | **92.1** | **97.6** | **90.4 (+2.4)** |
| **Motion Blur** | | | | | |
| Task Arithmetic | 65.3 | 68.1 | 80.0 | 64.2 | 69.4 |
| AdaMerging | 71.2 | 74.6 | 82.7 | 94.1 | 80.6 |
| *StatsMerging* | **73.5** | **76.9** | **89.2** | **95.2** | **83.7 (+3.1)** |
| **Impulse Noise** | | | | | |
| Task Arithmetic | 62.1 | 49.1 | 72.7 | 40.4 | 56.1 |
| AdaMerging | 67.2 | 30.8 | 75.9 | 77.5 | 62.8 |
| *StatsMerging* | **70.4** | **50.4** | **77.6** | **78.1** | **69.1 (+6.3)** |
| **Gaussian Noise** | | | | | |
| Task Arithmetic | 63.6 | 55.4 | 75.9 | 49.4 | 61.1 |
| AdaMerging | 69.9 | 41.2 | 80.6 | 76.0 | 66.9 |
| *StatsMerging* | **71.2** | **53.6** | **82.1** | **78.0** | **71.2 (+4.3)** |

**Input Noise Tolerance Boundary.** To test the boundry of input noise tolerance, we conducted experiments on merging two vision tasks on RESISC45 (RE) and EuroSAT (EU) on images with three levels of Gaussian noise: **Low noise** ($\sigma = 10$), **Medium noise** ($\sigma = 15$), and **High noise** ($\sigma = 20$). Results are shown in Table 11. In summary, as the noise level increased, the performance of both methods degraded. However, our proposed method *StatsMerging* consistently achieved higher accuracy than AdaMerging++ across all levels of Gaussian noise.

Table 11: Comparison of *StatsMerging* and AdaMerging on two vision tasks RESISC45 (RE) and EuroSAT (EU)) under three Gaussian noise levels (Low, Medium, and High). Numbers represent Avg Acc (%) across two tasks.

| Method | Low | Medium | High |
|---|---|---|---|
| AdaMerging++ | 56.0 | 48.4 | 35.9 |
| *StatsMerging* | **57.3 (+1.3)** | **50.1 (+1.7)** | **36.7 (+0.8)** |

**Label Noise Tolerance.** We use the entropy of a task expert's prediction (a model fine-tuned on task k) based on its output probability distribution, we further normalized the entropy values to the range $[0, 1]$ and split the dataset according to three noise levels evenly based on the normalized entropy: **Low noise** $[0, 0.33]$, **Medium noise** $[0.33, 0.66]$, and **High noise** $[0.66, 1]$, where numbers represent the boundaries of the normalized entropy. We note that entropy computed in this way represents the

confidence of a model, particularly the Task-Specific Teacher, acting as a proxy for label noise level, e.g., lower entropy indicates higher model confidence, and thus lower label noise, and vice versa. This interpretation aligns with its usage in the literature.

We conducted experiments on eight vision tasks under three noise levels. Results are shown in Table 12. We summarize the **new key insights** as follows: both methods achieved their best performance on low-noise labels (as expected) with over $95\%$ Avg Acc, and gradually degraded to around $80\%$ as the noise level increased. Our proposed method, *StatsMerging*, consistently outperformed AdaMerging++, with performance gains of $+4.4\%$, $+6.2\%$, and $+8.5\%$ under low, medium, and high noise levels, respectively. Both methods appear to be learnable across three noise levels. We did not observe any noise level boundary where *StatsMerging* underperformed AdaMerging++.

Table 12: Comparison of *StatsMerging* and AdaMerging on eight vision tasks under three noise levels (Low, Medium, and High). Numbers represent Avg Acc (%) across eight tasks.

| Method | Low | Medium | High |
|---|---|---|---|
| AdaMerging++ | 95.5 | 91.4 | 80.1 |
| *StatsMerging* | **99.9 (+4.4)** | **97.6 (+6.2)** | **88.6 (+8.5)** |

### A.5.4 EFFECT OF MODEL CAPACITY ON SML

Table 13 evaluates StatsMergeLearner (SML) with different design choices across RESISC45 and EuroSAT. While increasing capacity with deeper MLPs or a lightweight Transformer provides marginal accuracy gains (+0.1 - 0.3) and faster convergence, these come at the cost of higher parameter counts and computational complexity. The 2-layer MLP strikes a favorable balance between accuracy and efficiency, preserving the lightweight nature of SML while demonstrating the effectiveness of the overall framework.

Table 13: Avg Acc (%) performance of SML with different capacities on RE and EU. *: Current capacity in the submission. L: Layer.

| SML Design Choice | RE | EU | Avg Acc | #Params (M) | MACs (M) | FLOPs (M) |
|---|---|---|---|---|---|---|
| Individual Model | 96.1 | 99.7 | 97.9 | – | – | – |
| 2L MLP* | 96.0 | 98.0 | 97.4 | 0.366 | 0.73 | 1.46 |
| 4L MLP | 97.0 | 98.0 | 97.5 (+0.1) | 0.732 | 1.46 | 2.91 |
| 2L Transformer | 97.0 | 98.5 | 97.7 (+0.3) | 0.396 | 0.79 | 1.58 |

### A.5.5 LABEL TYPE AND LOSS FUNCTION ANALYSIS

In this section, we analyze the performance of training *StatsMergeLearner* on two types of pseudo labels: (1) Soft Pseudo Labels, and (2) Hard Pseudo Labels, the former of which is commonly employed in knowledge distillation frameworks (Gou et al., 2021; Hinton et al., 2015) especially for classification tasks. Formally, we present two versions of our training losses:

**Soft Pseudo Labels (SPL):** The predicted class probability distribution. Thus we use Kullback–Leibler divergence (KL-Div) (Kullback & Leibler, 1951) loss function:

$$\mathcal{L}_{\text{KL}} = \sum_{c=1}^{C_m} p_{c,k} \log\left(\frac{p_{c,k}}{q_c}\right) \tag{32}$$

where $p_{c,k}$ is the predicted probability of class $c$ from the pre-trained model $\theta_k$ on task $k$, and $q_c$ is the predicted probability of class $c$ from the merged model $\theta_m$.

**Hard Pseudo Labels (HPL):** The predicted class label in one-hot encoded format. Therefore, the cross-entropy loss is applied:

$$\mathcal{L}_{\text{CE}} = -\sum_{c=1}^{C_m} \hat{y}_{c,k} \log(\hat{y}_c)) \tag{33}$$

Results are shown in 14. We highlight two key observations: (1) Training *StatsMergeLearner* with Hard Pseudo Labels (HPL) using cross-entropy loss (KD CE) yields performance comparable to training with ground-truth labels (GT CE), achieving $81.2\%$ vs. $88.5\%$ at the task-wise (TW) level and $83.5\%$ vs. $90.4\%$ at the layer-wise (LW) level. Importantly, *StatsMerging* eliminates the need for manually annotated labels, validating our intuition of leveraging task-specific teacher knowledge for supervision. (2) When trained on Soft Pseudo Labels (SPL) using KL-Divergence loss (KL-Div), *StatsMergeLearner* underperforms relative to HPL with cross-entropy, obtaining $73.3\%$ vs. $81.2\%$ at the TW level and $52.4\%$ vs. $83.5\%$ at the LW level, respectively.

We hypothesize that the observed performance drop is due to noisy inter-class relationships within the aggregated dataset (Yuan et al., 2021). While a detailed investigation of these relationships is beyong the scope of this work on model merging, we believe it presents promising directions for future research.

**Label & Loss Function Study.** We conduct a loss function study on ViT-B/32 (4) models merged from four tasks, as shown in Table 4. Observe that *StatsMerging* trained on pseudo labels via Task-Specific Teacher Distillation (KD) achieves similar performance to *StatsMerging* trained on ground-truth labels (GT), with $88.5\%$ and $81.2\%$ average accuracy in TW and $90.4\%$ and $83.5\%$ in LW levels.

Table 4. Multi-task performance (Avg Acc %) of *StatsMerging* when merging ViT-B/32 (4) models across four tasks. *StatsMerging* shaded in gray. GT: Ground Truth. KD: Knowledge Distillation. TW: Task-wise. LW: Layer-wise.

| Loss | Level | CA | EU | RE | GT | Avg Acc |
|------|-------|------|------|------|------|---------|
| GT | TW | 73.2 | 94.2 | 91.1 | 95.6 | 88.5 |
| KD | TW | 64.2 | 88.6 | 85.2 | 86.7 | 81.2 |
| GT | LW | 75.6 | 96.3 | 92.1 | 97.6 | 90.4 |
| KD | LW | 68.7 | 91.6 | 87.2 | 93.5 | 83.5 |

Table 14: Multi-task performance (Avg Acc %) of *StatsMerging* when merging ViT-B/32 (4) models on four tasks. *StatsMerging*: shaded in gray. GT: Ground Truth. KD: Knowledge Distillation. GL: Granularity level. TW: Task-wise. LW: Layer-wise.

| GL | Loss | CA | EU | RE | GT | Avg Acc |
|----|----------|------|------|------|------|---------|
| TW | GT CE | 73.2 | 94.2 | 91.1 | 95.6 | 88.5 |
| TW | KD KL-Div | 56.5 | 97.6 | 56.5 | 82.4 | 73.3 |
| TW | KD CE | 64.2 | 88.6 | 85.2 | 86.7 | 81.2 |
| LW | GT CE | 75.6 | 96.3 | 92.1 | 97.6 | 90.4 |
| LW | KD KL-Div | 53.1 | 41.4 | 65.9 | 49.1 | 52.4 |
| LW | KD CE | 68.7 | 91.6 | 87.2 | 93.5 | 83.5 |

A.5.6   IMPACT OF DATA SIZE

The performance gain of *StatsMerging++* with more validation data is much larger than that of AdaMerging++ as shown in Table 15. When the data rate increases from 1% to 5%, LW *StatsMerging++* improves from 84.5% to 94.5%, whereas LW AdaMerging++ only increases from 80.6% to 81.0%. This demonstrates LW *StatsMerging++* is more data efficient than LW AdaMerging++.

Table 15: Impact of the amount of available data on performance (Avg Acc %) when merging ViT-B/32 models. *StatsMerging* is shaded in gray.

| Method | Data | SU | CA | RE | EU | SV | GT | MN | DT | Avg Acc |
|---|---|---|---|---|---|---|---|---|---|---|
| LW AdaMerging | 1% | 61.9 | 66.3 | 81.8 | 86.0 | 88.6 | 85.8 | 97.4 | 52.5 | 77.5 |
| LW AdaMerging++ | 1% | 66.9 | 68.6 | 81.4 | 91.8 | 89.2 | 87.1 | 98.1 | 61.8 | 80.6 |
| **LW StatsMerging** | 1% | 67.4 | 74.1 | 82.9 | 91.1 | 89.8 | 94.7 | 98.3 | 77.5 | 84.5 |
| LW AdaMerging | 5% | 63.7 | 68.6 | 79.1 | 93.3 | 86.5 | 91.7 | 97.2 | 61.9 | 80.1 |
| LW AdaMerging++ | 5% | 66.4 | 68.4 | 81.5 | 92.9 | 90.0 | 89.0 | 98.2 | 61.5 | 81.0 |
| **LW StatsMerging++** | 5% | **92.4** | **95.4** | **95.1** | 92.9 | **94.6** | 98.7 | **98.5** | **88.4** | **94.5 (+5.1)** |
| LW AdaMerging | 100% | 64.5 | 68.1 | 79.2 | 93.8 | 87.0 | 91.9 | 97.5 | 59.1 | 80.1 |
| LW AdaMerging++ | 100% | 66.6 | 68.3 | 82.2 | 94.2 | 89.6 | 89.0 | 98.3 | 60.6 | 81.1 |

**Difference between StatsMerging++ and AdaMerging++**:

- *StatsMerging++* uses 5% validation data instead of the 1% used in StatsMerging.
- **AdaMerging++** additionally removes parameter redundancies and resolves sign conflicts via TIES-Merging; this modification is independent of the amount of data used.

Table 16: Impact of Training Size on LW *StatsMerging* for Eight Vision Merging Tasks (Avg Acc %).

| Training Data Percentage | 5% | 25% | 50% | 75% |
|---|---|---|---|---|
| **Avg Acc** | 84.50 | 86.60 | 89.08 | 89.14 |

### A.5.7 TRAINING CURVE

We present the training curve of ViT-B/32 across eight Vision tasks in Fig. 8. The sharp drop in learning rate at around step 420 stabilizes the *StatsMergeLearner* updates, reduces gradient noise, and allows the merged model to settle into a flatter minimum. This scheduling effect explains the sudden increase in training accuracy across all eight tasks, rather than any change in the data or validation split. This behavior is well-known in deep neural network training and is consistent with empirical and theoretical evidence in prior work (Luo et al., 2019; Ren et al., 2024)

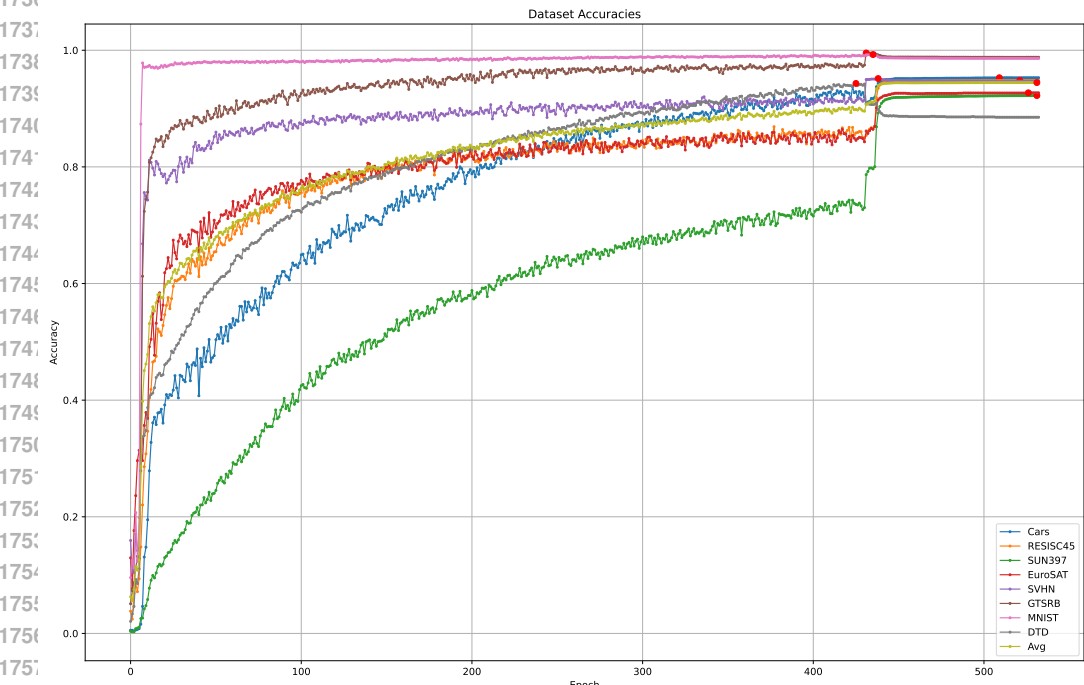

Figure 8: *StatsMerging++* Training Accuracy Curve.

## A.6 EXTENDED RELATED WORK

**Model Merging Foundations.** Recent efforts in model merging have introduced various strategies to efficiently combine multiple models without retraining. Approaches such as ZipIt (Zhang et al., 2024a), EMR-Merging (Huang et al., 2024), and Training-Free Pre-trained Model Merging methods (Sun et al., 2025; Chen et al., 2024) emphasize data-free, tuning-free methodologies, often leveraging weight-space heuristics or task-vector alignment. Techniques like Pareto Merging (Chen & Kwok, 2025), MAP (Li et al., 2024), and $C^2 M^3$ (Crisostomi et al., 2024) formulate model merging as a multi-objective or constrained optimization problem to preserve task performance across domains. Other works such as Parameter Competition Balancing (Guodong et al.) and Sharpness-Aware Fine-Tuning (Lee et al., 2025) address parameter interference during merging. Meanwhile, methods like LayerMerge (Kim et al., 2024) and MERGE3 (Mencattini et al., 2025) aim to improve scalability and computational efficiency, making merging feasible on consumer-grade hardware.

**Merging Methods in Computer Vision.** The application of model merging techniques in computer vision is relatively less explored compared to natural language processing (Yadav et al., 2023b; Ilharco et al., 2023). Computer vision models, particularly deep convolutional neural networks (CNNs) (Krizhevsky et al., 2012; He et al., 2016; Simonyan & Zisserman, 2014) and Vision Transformers (ViTs) (Dosovitskiy et al., 2021a; Touvron et al., 2021), learn complex, hierarchical feature representations that are highly sensitive to task-specific optimizations (Izmailov et al., 2018). Simple averaging techniques often fail due to the non-convex nature of the loss landscape and the divergence of learned feature spaces across different visual tasks. Recent advancements (Matena & Raffel, 2022; Yang et al., 2023) have shown potential, but often lack explicit mechanisms to account for the unique properties inherent in visual data and architectures, such as spatial relationships in CNNs or attention mechanisms in ViTs. Furthermore, the effectiveness of these methods across the broad spectrum of computer vision tasks, including low-level restoration (Zhang et al., 2017; Saharia et al., 2022), mid-level detection (Ren et al., 2015; Carion et al., 2020b), and high-level classification (He et al., 2016), has not been comprehensively validated. Our work addresses these limitations by introducing a novel merging framework that leverages internal model weight statistics to guide the merging process, making it more adaptable and effective across diverse computer vision tasks and architectures.

**Relationship to KnOTS.** *Compare KnOTS and combine it with the proposed SML.*

We included KnOTS (Stoica et al., 2024) as an additional baseline and evaluated *StatsMerging* + KnOTS. As shown in Table 17, *StatsMerging* + KnOTS performs worse than our proposed *StatsMerging* in this two-task setting. We hypothesize that this is due to (i) KnOTS being sensitive to SVD rank selection and scaling, and (ii) its design being more beneficial for larger and more diverse task sets. Although KnOTS converges faster, it incurs approximately $10\times$ higher training cost per epoch due to repeated SVD computations.

Table 17: Comparison of different merging methods on two-task merging (RE, EU).

| Method | RE | EU | Avg Acc (%) |
|---|---|---|---|
| Individual | 96.1 | 99.7 | 97.9 |
| Task Arithmetic | 85.2 | 96.7 | 90.9 |
| TIES-Merging | 86.4 | 97.2 | 91.8 |
| *StatsMerging* + KnOTS | 92.1 | 94.2 | 93.2 |
| ***StatsMerging*** | **96.0** | **98.0** | **97.4 (+4.2)** |

**Relationship to LiNeS.** *Similarity:* Both share a similar goal of preserving common features across tasks while retaining task-specific representations. *Difference:* LiNeS (Wang et al., 2024a) scales the updates from shallow to deep layers linearly, controlled by $\alpha$ and $\beta$. In Layer-Wise (LW) *StatsMerging*, merging coefficients ($\lambda$) are optimized across the entire merged model by SML. Therefore, in theory, $\lambda$ should jointly account for the scales of updates from shallow to deeper layers. In addition, SML does not assume the linear scaling from shallow to deeper layers as in LiNeS.

We therefore posit that SML (and other learning-based methods) may not benefit significantly from directly integrating LiNeS scaling coefficients, either during training or in post-training stages. This is consistent with the fact that in the LiNeS paper, the merging methods that LiNeS integrates with are heuristic-based, including Task Arithmetic, Ties-Merging, Consensus Merging (Table 18), and

Model Soup. The only learning-based method reported in the experiments is AdaMerging, which was only used solely for comparison, if I am not mistaken. Although SML can be combined with LiNeS in practice/implementation, we find it theoretically unnecessary.

*Comparison:* We present the comparison of LiNeS and our updated *StatsMerging* (w SML) on merging ViT-B/32 in Table 18. Our proposed StatsMerging (84.5%) significantly outperforms the best reported LiNeS result (77.2%).

Table 18: Multi-task merging performance (Avg Acc %) when merging ViT-B/32 models on eight tasks. Results of our method *StatsMerging* are in bold. LW: Layer-wise.

| Method | Avg Acc |
|---|---|
| Task Arithmetic | 69.7 |
| Task Arithmetic + LiNeS | 74.2 |
| Ties-Merging | 73.6 |
| Ties-Merging + LiNeS | 77.2 |
| Consensus Merging | 74.5 |
| Consensus Merging + LiNeS | 77.6 |
| LW AdaMerging | 80.1 |
| LW AdaMerging++ | 81.1 |
| **LW *StatsMerging*** | **84.5** |

### A.7 VISUAL ASSETS ATTRIBUTION

We credit the guru (Task-Specific Teachers) and student visual icons to Freepik–Flaticon (https://www.flaticon.com/free-icons/idea), which enhance the clarity and presentation quality of our approach.

### A.8 FUTURE WORK AND LIMITATIONS

In this work, we focus on vision-based classification and simple NLP tasks, leaving extensions to other domains, such as object detection (Tan et al., 2020), super-resolution (Sun et al., 2022), and image and video restoration (Liang et al., 2021; Merugu et al., 2025), for future work. Additionally, expanding this approach to beyond vision and language tasks, particularly large language models (LLMs) (Yang et al., 2024; Song et al., 2024; Zhang et al., 2024b; Tie et al., 2025; Kallini et al., 2025), as well as to multi-modal learning (Zhu et al., 2025; Du et al., 2025; Bousselham et al., 2024; Lin et al., 2024), represents a promising direction for further research. Moreover, we identify a direction for future work that can facilitate more efficient SML learning. While our work primarily focuses on empirical results, we regard theoretical development, such as formal proofs, as an important direction for future research.

