# OpenReview forum: "StatsMerging: Statistics-Guided Model Merging via Task-Specific Teacher Distillation"
_ICLR.cc/2026/Conference — Submitted to ICLR 2026_

### Official Review · Reviewer_kzYd · 2025-10-26

**Soundness:** 3
**Presentation:** 2
**Contribution:** 2
**Rating:** 4
**Confidence:** 4

**Summary:**

The paper proposes StatsMerging, a method for merging pre-trained models using weight distribution statistics to guide the learning of merging coefficients.

**Strengths:**

1. The use of SVD singular values as a proxy for task/layer importance is a reasonable extension of existing merging techniques. This is also used in some recently published model merging methods, such as TSVM (task singular vector merging).
2. Handling heterogeneous architectures via distill-then-merge is a practical contribution.

**Weaknesses:**

Below are some weaknesses of the manuscript and suggestions for improve the manuscript:

1. The averaged performance of individual fine-tuned models could be shown in Figure 4 using a vertical line.
2. Lack comparison with recent state-of-the-art training-free model merging methods, such as TSVM (task singular vector merging) and RegMean++.
3. Mirror typos:
    - At line 371 on page 7, should "MEMoE" be "WEMoE"?
    -  At line 226-235 on page 5, please verify that all instances of the symbols $\sigma_{r}$ and $\sigma’_{r}$ are used correctly.
4. SVD rank is fixed at 3 without ablation—why not 1, 5, or 10? How sensitive is performance to rank choice?
5. Weight statistics are used but no justification for why these capture "task importance" better than alternatives like Fisher information used in Fisher merging.

**Questions:**

1.Why does the accuracy increase so dramatically at about 420 steps as shown in Figure 8?

---

> ### Author Response · Authors · 2025-11-21
> **Response to Reviewer kzYd (R4) - Weaknesses**
>
> We thank Reviewer __kzYd__ (R4) for (S1) acknowledging our use of SVD singular values as a proxy for task/layer importance and (S2) recognizing the practical contribution of our distill-then-merge strategy for merging models with heterogeneous architectures.
>
> We also appreciate Reviewer __kzYd__(R4) for their time and effort in providing a thoughtful review to help improve our work.
>
> To help navigate our responses, we use the following notation: __A__ (Answer), __R__ (Reviewer), __W__ (Weakness), __Q__ (Question), and __T__ (Table). Responses addressing weaknesses are labeled __AR4W#__, and those addressing questions are labeled __AR4Q#__.
>
> While we aim to address all concerns as thoroughly as possible within this rebuttal, some questions may be clarified further during the discussion period. We appreciate your patience and engagement.
>
> ---
>
> > __R4W1__: The averaged performance of individual fine-tuned models could be shown in Figure 4 using a vertical line.
>
> __AR4W1__: We appreciate the reviewer's suggestion in enhancing the figure presentation. The new figure with a vertical line representing individual fine-tuned models will be added in the updated draft.
>
> ---
>
> > __R4W2__: Lack comparison with recent state-of-the-art training-free model merging methods, such as TSVM (task singular vector merging) and RegMean++.
>
> __AR4W2__: Please refer to Table __AR1W2T1__. Overall, StatsMerging++ still achieves the best performance, with the only close case being ViT-L/14, where TSV-M reaches 94.1% compared to our 94.7%.
>
> We have added this result and the corresponding related works to the updated draft. In practice, it is challenging to include every new method due to the rapid emergence of model merging approaches; however, we have made our best effort to incorporate the most relevant works available at submission time. We sincerely appreciate the reviewer’s efforts in pointing out recent developments, which helps us better position our work within the current literature.
>
> ---
>
> > __R4W3__: Mirror typos.
> > __R4W3.1__: At line 371 on page 7, should "MEMoE" be "WEMoE"?
>
> __AR4W3__: We thank the reviewer for their careful attention to detail, which helped us identify and correct typographical errors.
>
> __AR4W3.1__: __Yes__. Will be corrected in the updated draft. As humans, we may occasionally overlook minor typos, and we appreciate the reviewer’s help in catching them.
>
>
> > __R4W3.2__: At line 226-235 on page 5, please verify that all instances of the symbols $\sigma_{r}$ and $\sigma_{r}'$ are used correctly.
>
> __AR4W3.2__: We have verified that they are __correct__. We quote the definitions in the __original submission__ for your reference:
>
> Line 219: variance $\sigma^2=Var(\theta_{k})$
>
> where $\sigma$ represents the standard deviation (Std Dev).
>
> Line 228: singular value vector given rank r: **$\sigma_{r}$'** =  [$\sigma_{1}$', $\sigma_{2}$', ..., $\sigma_{r}$'].
>
> We intentionally use $\sigma_{r}$' (with a prime "$'$") instead of its original convention $\sigma_{r}$ to avoid confusion with the standard deviation, which is also commonly denoted by $\sigma$.
>
> ---
>
> > __R4W4__: SVD rank is fixed at 3 without ablation—why not 1, 5, or 10? How sensitive is performance to rank choice?
>
> __AR4W4__: Please refer to Table __AR3W1T1__ in __AR3W1, AR3Q1__. StatsMerging++ achieves the best performance with Rank 3 compared to other ranks.

---

> ### Author Response · Authors · 2025-11-22
> **Response to Reviewer kzYd (R4) - Questions**
>
> > __R4Q1__: Why does the accuracy increase so dramatically at about 420 steps as shown in Figure 8?
>
> __AR4Q1__:
>
> The sharp drop in learning rate at around step 420 stabilizes the StatsMergeLearner’s updates, reduces gradient noise, and allows the merged model to settle into a flatter minimum. This scheduling effect explains the sudden increase in training accuracy across all eight tasks, rather than any change in the data or validation split. This behavior is well-known in deep neural network training (not specific to training-based model merging methods) and is consistent with empirical and theoretical evidence [6] (Figure 3: Training and test accuracy for DenseNet-121 and ResNet-34 on CIFAR-10) in prior work [7]. We will incorporate this analysis and citations [6, 7] into the revised draft.
>
> [6] Luo, Liangchen, Yuanhao Xiong, Yan Liu, and Xu Sun. "Adaptive Gradient Methods with Dynamic Bound of Learning Rate." In International Conference on Learning Representations. ICLR, 2019.
>
> [7] Ren, Yinuo, Chao Ma, and Lexing Ying. "Understanding the generalization benefits of late learning rate decay." In International Conference on Artificial Intelligence and Statistics, pp. 4465-4473. PMLR, 2024.

---

> ### Author Response · Authors · 2025-12-03
> **Response to Reviewer kzYd (R4) - Weaknesses**
>
> > __R4W5__: Weight statistics are used but no justification for why these capture "task importance" better than alternatives like Fisher information used in Fisher merging.
>
> __AR4W5__: We show that mean and variance determine the second-moment in Sec. A.4.2.
>
> For a fine-tuned weight $\theta$, define
> $$ \mu = E[\theta] $$
> $$ \sigma^2 = Var(\theta) = E[(\theta − \mu)^2] $$
>
> $$E[\theta²]
>  = E[(\theta − \mu + \mu)^2]
>  = E[(\theta − \mu)^2] + 2\mu E[\theta − \mu] + \mu^{2}
>  = \sigma^{2} + \mu^{2}                                         $$
> Thus
> $$ E[\theta^{2}] = Var(\theta) + (E[\theta])^{2} = \sigma^{2} + \mu^{2} $$
>
> The diagonal Fisher information provides a complementary second-order quantity, approximating local curvature via the expected squared gradient
> $$ F_i ≈ E[ ( \partial log (p) / \partial\theta_i )^{2} ]$$
>
> Considering these together, Fisher information estimates local second-order sensitivity via squared gradients, but requires many forward/backward passes and is sensitive to gradient noise. Our mean and variance offer global, data-free second-moment statistics of the weights, which approximate the diagonal Fisher signal while being dramatically more stable and efficient.
>
> We further explain the rationale behind our statistical feature selection compared to prior works in the "Motivation" subsection in Sec. 3.2 WEIGHT STATISTICS-GUIDED MODEL MERGING highlighted in teal in the updated draft.

---

### Official Review · Reviewer_RxsX · 2025-10-27

**Soundness:** 2
**Presentation:** 2
**Contribution:** 3
**Rating:** 6
**Confidence:** 4

**Summary:**

This paper introduces StatsMerging, a statistics-guided model merging approach designed to consolidate multiple task-specific deep models—primarily in vision and NLP domains—without requiring ground truth labels or access to test samples. The architecture leverages Singular Value Decomposition (SVD)-derived statistics (mean, variance, norm, top singular values) from model weights and trains a lightweight StatsMergeLearner (SML) to predict adaptive merging coefficients through a novel task-specific teacher distillation process. The method extends to heterogeneous architectures and is validated across eight vision and seven NLP tasks, showing improvements over several baselines in average accuracy and robustness to data/label noise.

**Strengths:**

- The article demonstrates the absolute advantages of this method across various task sets and datasets through extensive experiments.
- The method proposed in the article does not require manually annotated labels; instead, it leverages pseudo-labels generated by existing models. This eliminates the need for manual annotation and provides certain support for the subsequent expansion of different scenarios.
- The structural design of SML is very simple with low training costs, and its computational cost is much lower compared to other model merging methods.

**Weaknesses:**

- There is a lack of certain theoretical or experimental explanations. For instance, it does not clarify the reason for choosing rank=3 when performing SVD.
- The experimental details are insufficient. For example, it fails to specify the exact amount of data used for training StatsMerging and StatsMerging++ respectively.
- There may be issues with the experimental results. For instance, the results of LW StatsMerging++ in Table 3 and Table 7 of the Appendix are inconsistent.

**Questions:**

- Why is a rank of 3 chosen for SVD? Is it necessary to add ablation experiments on rank to identify the tradeoff between the overhead caused by rank and performance?
- Can a graph showing the relationship between the amount of training data and performance be provided? Additionally, can the specific amounts of data used for StatsMerging and StatsMerging++ be confirmed?
- Can the reason for the inconsistent results of LW StatsMerging++ in Table 3 and Table 7 of the Appendix be clearly explained? Alternatively, can the corrected experimental results be provided after re-evaluation?

---

> ### Author Response · Authors · 2025-11-21
> **Response to Reviewer RxsX (R3)**
>
> We thank Reviewer __RxsX__ (R3) for (S1) acknowledging our absolute advantages over baselines through extensive experiments, (S2) recognizing that our method eliminates the need for manual annotation, and (S3) highlighting the lightweight, low-cost design of SML.
>
> We also appreciate Reviewer __RxsX__ (R3) for their time and effort in providing a thoughtful review to help improve our work.
>
> To help navigate our responses, we use the following notation: __A__ (Answer), __R__ (Reviewer), __W__ (Weakness), __Q__ (Question), and __T__ (Table). Responses addressing weaknesses are labeled __AR3W#__, and those addressing questions are labeled __AR3Q#__.
>
> While we aim to address all concerns as thoroughly as possible within this rebuttal, some questions may be clarified further during the discussion period. We appreciate your patience and engagement.
>
> ---
>
> > __R3W1, R3Q1__: SVD Rank 3 Choice Justification
>
> __AR3W1, AR3Q1__: Please refer to the results in Table __AR3W1T1__, which demonstrate that StatsMerging++ achieves the best performance with Rank 3 compared to others. Rank 3 captures around 95% energy while including higher-rank components (e.g., >3) mainly introduces noise and degrades performance. We will add these results to the revised draft and thank Reviewer RxsX (R3) for helping make our work more comprehensive.
>
> **Table AR3W1T1: Impact of Rank on Multi-task merging performance (Avg Acc %) when merging StatsMerging++ ViT-B/32 models on eight
> vision tasks.**
> | Rank | Avg Acc (%)  |
> |-|-|
> | 1 | 86.5 |
> | 2 | 87.2 |
> | **3** | **94.5** |
> | 4 | 89.2 |
> | 5 | 86.7 |
> | 8 | 84.2 |
>
> For weight statistical feature justification please kindly refer to __AR2W1: Empirical evidence__.
>
> ---
>
> > __R3W3, R3Q3__: results of LW StatsMerging++ in Table 3 and Table 7 of the Appendix.
>
> __AR3W3, AR3Q3__:
>
> StatsMerging++
>
> Table 3: 94.5% Avg Acc trained for 500 epochs.
>
> Table 7: 89.0% Avg Acc trained for 389 epochs.
>
> Table 7 is to highlight that our method already achieves strong performance with an average accuracy of 89% trained after only 389 epochs, requiring fewer training epochs than methods such as AdaMerging.
>
> We verified that these results are correct, aligned with Figure 8: StatsMerging++ Training Accuracy Curve in the original draft, where x-axis shows training epochs. We will use 94.5% to be consistent in the updated draft.
>
> We thank Reviewer RxsX for the attention to such great details to help reduce potential errors of our work.

---

> > ### Comment · Reviewer_RxsX · 2025-11-24
> > **My doubts about R3Q2 and R3Q3**
> >
> > Thank you for your response. R3Q1 is now clear to me. However, R3Q2 was not addressed, and I remain uncertain about R3Q3. Specifically, the reported 94.5 result already surpasses both Individual and Traditional MTL models, which seems somewhat surprising. I also noted the authors’ Figure 8, but I am curious why there is a sharp performance surge around epoch 420. Does this phenomenon not appear in the other models?

---

> ### Author Response · Authors · 2025-11-25
> **Response to R3Q2 and R3Q3 - Follow-Up**
>
> Thank you for your questions and notified us that __R3Q1__ is __clear__.
>
> A: Answer. F: Follow-up.
>
> For __R3Q2__ please refer to __AR3Q2.1__ and __AR3Q2.2__.
>
> > __AR3Q3__: why there is a sharp performance surge around epoch 420.
>
>  __AR3Q3__: We posit that __SML converges faster after learning decayed at around Epoch 420__ managed by the learning rate (LR) Schedule. The learning rate at around Epoch 420 could be suitable for SML's weights to __reach the areas near the minimum (i.e, the optimal merging coefficients) faster__. SML's learning is affected by the learning rate dynamics, small or large LR may lead to ineffective or unstable learning. In our case, this behavior arises from the adaptive nature of the Adam optimizer in the implementation. This aligns with the observations in other ML/DL training pipelines in prior work [6] [7]. Empirically, we observe a relative sharp increase for at epoch 420 for this specific training. Please also refer to __AR4Q1__ (simply search the keyword "AR4Q1" on the page to navigate to the answer).
>
> > __R3Q3F1__: Does this phenomenon not appear in the other models?
>
> __AR3Q3F1__: It __appears__ in __many other models__ as well. In our experience, we empirically observe that sudden sharp increase happens not just in the model merging context but also in many other ML/DL training pipelines. For [6], we refer to the similar observation but for image classification (sharp increase at Epoch 150 in their case), specically _Figure 3: Training and test accuracy for DenseNet-121 and ResNet-34 on CIFAR-10_.
>
> As such (sharp performance increase in general ML/DL trainings), we did not originally include a detailed discussion of this sharp performance increase in our model merging paper. However, we will incorporate this discussion of this behavior while citing previous works [6, 7] in the revised draft and relate it to similar observations reported in prior work.
>
> [6] Luo, Liangchen, Yuanhao Xiong, Yan Liu, and Xu Sun. "Adaptive Gradient Methods with Dynamic Bound of Learning Rate." In International Conference on Learning Representations. ICLR, 2019.
>
> [7] Ren, Yinuo, Chao Ma, and Lexing Ying. "Understanding the generalization benefits of late learning rate decay." In International Conference on Artificial Intelligence and Statistics, pp. 4465-4473. PMLR, 2024.
>
> Lastly, we appreciate Reviewer __RxsX__ (R3)'s intriguing questions to help understand the deeper aspect of SML learning dynamics. Let us know if you have any further questions.

---

> ### Author Response · Authors · 2025-12-03
> **Response to Reviewer RxsX (R3)**
>
> > __R3W2__ & __R3Q2.2__: the exact amount of data used for training StatsMerging and StatsMerging++ respectively.
>
> __AR3W2__ & __AR3Q2.2__: 319,665 for the 8-task vision benchmark. For the effect of validation data please refer to __AR1Q1.1__ (results with 1%, 5%, and 100%).
>
> The exact number of data is detailed in Sec. A.1 EXPERIMENT SETTINGS for both vision and NLP benchmarks and Train/Val/Test Split 8:1:1 in Sec. A.3 DETAILS OF TASK-SPECIFIC TEACHER DISTILLATION in the __original submission__.
>
> Take the vision benchmark for example, the number of training samples is
> $$ 108,754 \times 0.8 (SU) + 8093 (CA) + 31,500 \times 0.8 (RE) + 27,000 \times 0.8 (EU) + 73,257 (SV) + 50,000 \times 0.8 (GT) + 60,000 (MN) + 5,640 \times 0.8 (GT) = 319,665 $$
>
> ---
>
> > __R3Q2.1__: the relationship between the amount of training data and performance.
>
> __AR3Q2.1__: The relationship is shown in Table __AR3Q2T1__ with 0% validation data used (no adaptation).
>
> **Table AR3Q2T1: Impact of Training Size on LW StatsMerging for Eight Vision Merging Tasks (Avg Acc %)**
> | Training Data Percentage          | Avg Acc |
> |----------------|----------|
> | 5% | 84.50 |
> | 25% | 86.60 |
> | 50% | 89.08 |
> | 75% | 89.14 |

---

### Official Review · Reviewer_5Rvs · 2025-10-31

**Soundness:** 2
**Presentation:** 2
**Contribution:** 2
**Rating:** 2
**Confidence:** 4

**Summary:**

This paper proposes a lightweight, learning-based model merging method named StatsMerging. The core idea is to adaptively predict the merging coefficients based on the weight distribution information.

**Strengths:**

1. The paper is well-organized and easy to follow.

2. The experiments demonstrate that the proposed method achieves promising results.

**Weaknesses:**

1. **Unclear motivation.** The usage of weight distributions plays a central role in the proposed method and serves as its main motivation. However, the paper lacks empirical or theoretical evidence to support the importance or effectiveness of this design choice.

2. **Limited technical contributions.** The use of knowledge distillation has been extensively studied in the context of model merging and other areas of machine learning. Similarly, learning adaptive merging coefficients has also been well explored in prior model merging research, which limits the novelty of the proposed approach.

3. **Reliance on training data.** The proposed method requires access to additional training data, whereas data-free model merging approaches have already been widely studied. This reliance may weaken the practical advantage of the proposed method.

**Questions:**

See weaknesses above.

**Details Of Ethics Concerns:**

No ethics concerns.

---

> ### Author Response · Authors · 2025-11-21
> **Response to Reviewer 5Rvs (R2) - Weaknesses**
>
> We thank Reviewer __5Rvs__ (R2) for (S1) recognizing the well-organized presentation of our work and (S2) acknowledging that our approach achieves promising results. We also appreciate their time and effort in providing the review to help improve our work.
>
> We also appreciate Reviewer __5Rvs__ (R2) for their time in providing the review to help improve our work.
>
> To help navigate our responses, we use the following notation: __A__ (Answer), __R__ (Reviewer), __W__ (Weakness), __Q__ (Question), and __T__ (Table). Responses addressing weaknesses are labeled __AR2W#__, and those addressing questions are labeled __AR2Q#__.
>
> While we aim to address all concerns as thoroughly as possible within this rebuttal, some questions may be clarified further during the discussion period. We appreciate your patience and engagement.
>
> ---
>
> > __R2W1__: Unclear motivation. The usage of weight distributions plays a central role in the proposed method and serves as its main motivation. However, the paper lacks empirical or theoretical evidence to support the importance or effectiveness of this design choice
>
> __AR2W1__: __Empirical evidence__: We quote Table 5 in Sec. 4.3 StatsMerging ANALYSIS - Statistical Feature Ablation Study from our __original submission__ for your reference:
>
> **Table 5:** Multi-task performance (Avg Acc %) of **StatsMerging** when ablating statistical features of ViT-B/32 (4) models on four tasks: CA, EU, RE & GT.  **Bold**: top score.
>
> **Same Architecture**:
> | $μ$ | σ² | m | σ′| **Avg Acc** |
> |-----------------------------|--------|--------|--------|-------------|
> | ✓ | | |  | 83.4 |
> | ✓ | ✓ |    |    | 84.1 (+0.7) |
> | ✓ | ✓ | ✓  |    | 87.2 (+3.1) |
> | **✓** | **✓** | **✓** | **✓** | **90.2 (+3.0)** |
>
> **Different Architecture**:
> | $μ$ | σ² | m | σ′ | **Avg Acc** |
> |-----------------------------|--------|--------|--------|-------------|
> | ✓ | | |  | 76.2 |
> | ✓ | ✓ |    |    | 77.5 (+1.3) |
> | ✓ | ✓ | ✓  |    | 78.1 (+0.6) |
> | **✓** | **✓** | **✓** | **✓** | **81.3 (+3.2)** |
>
> $\mu$: mean. $\sigma^{2}$: variance. $m=||\theta||$: magnitude of weight $\theta$. $\sigma'$: singular value (Rank 3 by default).
>
> Progressively adding statistical features of the pre-trained weights $\theta$ consistently improves merging performance, and using all of them yields the best results. We quote the __design choice__ conclusion in the __original submission__ in Line 454 for your reference: _"Results in Table 5 show that combining all statistical features improves merging performance, validating our __design choice__."_

---

> > ### Author Response · Authors · 2025-12-03
> > **Response to Reviewer 5Rvs (R2) - Weaknesses Follow-Up**
> >
> > F: Follow-up.
> >
> > > __R2W1__: Unclear motivation. The usage of weight distributions plays a central role in the proposed method and serves as its main motivation. However, the paper lacks empirical or theoretical evidence to support the importance or effectiveness of this design choice.
> >
> > __AR2W1F1__: __Motivation__. We added two paragraphs "Motivation" in Sec. 3.2 WEIGHT STATISTICS-GUIDED MODEL MERGING before formally introducing Weight Statistics and SML in teal in the updated draft. The use of weight statistics was inspired by several key observations in prior works. The effectiveness of combined use of weight statistics is evidenced in __AR2W1: Empirical evidence__.
> >
> > __Theoretical evidence__. We have included theoretical analyses in Sec. A.4 THEORETICAL ANALYSES highlighted in teal in the updated draft.

---

> ### Author Response · Authors · 2025-11-21
> **Response to Reviewer 5Rvs (R2) - Weaknesses**
>
> > __R2W2__: Limited technical contributions. The use of knowledge distillation has been extensively studied in the context of model merging and other areas of machine learning. Similarly, learning adaptive merging coefficients has also been well explored in prior model merging research, which limits the novelty of the proposed approach.
>
> __AR2W2__:
>
> We agree with Reviewer 5Rvs (R2) that knowledge distillation has been extensively studied in the machine learning community, and we explicitly acknowledge representative prior work in Line 264 of the __original submission__:
>
> "With the help of well-trained teachers, knowledge distillation (Hinton et al., 2015) has been proven as an effective
> way to train a model without human annotations."
>
> __Role of knowledge distillation in our work__:
>
> The use of knowledge distillation is not our objective. Instead, it serves two specific purposes: (1) to assist SML in learning task-specific expert __statistical weight distributions__ without human annotations, and (2) to enable __heterogeneous architectural merging__, as described in the second (C2) and third (C3) contributions in Line 099 of the __original submission__.
>
> __Relation to learning adaptive merging coefficients__:
>
> We would like to clarify that we do not claim learning adaptive merging coefficients itself as our contribution. Rather, we explicitly position our work as an extension of this line of research, with clear acknowledgement of prior methods in the main manuscript, particularly AdaMerging. Building on these foundations, our goal is to substantially improve merging performance, as summarized in the first (C1) and forth (C4) contributions (Line 97, Page 2) of the __original submission__:

---

> ### Author Response · Authors · 2025-11-21
> **Response to Reviewer 5Rvs (R2) - Weaknesses**
>
> > __R2W3__: Reliance on training data. The proposed method requires access to additional training data, whereas data-free model merging approaches have already been widely studied. This reliance may weaken the practical advantage of the proposed method.
>
> __AR2W3__:
>
> While our method uses unlabeled data, this is standard in recent high-performing merging approaches (e.g., AdaMerging, AdaMerging++). Data-free methods such as RegMean, TIES, or Fisher are valuable baselines but often struggle with heterogeneous architectures and complex multi-task settings. Our contribution focuses on a more challenging yet practically important regime where a small amount of unlabeled samples (1–5%) is available, which is consistent with many real-world applications such as video analytics or self-driving (e.g., license plate recognition, scene classification, object segmentation). In this setting, StatsMerging++ achieves strong or superior performance with lower overhead. For example, on ViT-B/32 (Vision), data-free RegMean and TIES-Merging achieve only 71.8% and 72.4% Avg Acc, respectively, whereas StatsMerging++ reaches 94.5%.

---

### Official Review · Reviewer_ksSD · 2025-11-01

**Soundness:** 2
**Presentation:** 3
**Contribution:** 2
**Rating:** 4
**Confidence:** 5

**Summary:**

The paper introduces StatsMergeLearner, a network that predicts scaling coefficients for model merging from statistical features of model weights, trained using pseudo-labels generated by teacher models. It also proposes a Merge+Distill framework to homogenise model architectures—a prerequisite for merging—and reports strong gains over prior work across vision and language tasks.

**Strengths:**

(1) String Performance: Consistently outperforms Adamerging and WEMoE on the reported benchmarks.

(2) Practical pipeline: The proposed Merge+Distill framework provides a workable path to architectural homogenisation, enabling broader applicability of merging.

**Weaknesses:**

(1) Limited task scale: Evaluation is restricted to the 8-task vision suite; it omits the widely used 14- and 20-task benchmarks from [1] on ViT-B/32, ViT-B/16, and ViT-L/14, which are important for understanding scaling with respect to task count and backbone size.

(2) Baselines: Missing comparisons to recent data-free merging methods such as Isotropic Merging[2], TSV-M[3], and KnOTS[4].

(3) Ablations: Task-level StatsMerging is introduced but not compared against layer-wise StatsMerging across settings, making it unclear what the difference in performance looks like.

References:

 [1] Wang, Ke, et al. "Localizing task information for improved model merging and compression." arXiv preprint arXiv:2405.07813 (2024).

[2] Marczak, D., Magistri, S., Cygert, S., Twardowski, B., Bagdanov, A. D., & van de Weijer, J. (2025). No task left behind: Isotropic model merging with common and task-specific subspaces. arXiv preprint arXiv:2502.04959.

[3] Gargiulo, A. A., Crisostomi, D., Bucarelli, M. S., Scardapane, S., Silvestri, F., & Rodola, E. (2025). Task singular vectors: Reducing task interference in model merging. In Proceedings of the Computer Vision and Pattern Recognition Conference (pp. 18695-18705).

[4] Stoica, G., Ramesh, P., Ecsedi, B., Choshen, L., & Hoffman, J. (2024). Model merging with svd to tie the knots. arXiv preprint arXiv:2410.19735.

**Questions:**

(1) Effect of extra validation data: The jump from LW-StatsMerging to LW-StatsMerging++ is substantial. Do other adaptation methods (e.g., Adamerging) show similar improvements as validation data increases? Additionally, since StatsMerging does not introduce any new hyperparameters of its own, what is the role of validation data?


(2) Data sources: What exact data are used for training StatsMergeLearner and for distillation? Is it the training samples from each task? Also, is the setup strictly unlabeled for both training and validation, or is any labelled data employed?


(3) Training cost: Line 332 states StatsMergeLearner is trained for 500 epochs, which seems heavy compared to, e.g., Adamerging’s ~500 steps, which is okay since it also outperforms it, but should not be regarded as a lightweight method. Do (a) the number of models being merged and (b) backbone size (e.g., ViT-L/14) influence the number of training steps used to train the StatsMergeLearner?

**Details Of Ethics Concerns:**

No concerns.

---

> ### Author Response · Authors · 2025-11-20
> **Response to Reviewer ksSD (R1) - Weaknesses**
>
> We thank Reviewer __ksSD__ (R1) for acknowledging __S1__ __consistent superior performance__ over Adamerging and WEMoE, as well as __S2__ acknowledging the practicality towards __architectural homogenisation merging__.
>
> We also appreciate Reviewer __ksSD__ (R1) for their thoughtful effort in providing feedback to improve our work, especially regarding the effect of task scaling, comparison to data-free merging methods, clarity with details.
>
> To help navigate our responses, we use the following notation: __A__ (Answer), __R__ (Reviewer), __W__ (Weakness), __Q__ (Question), and __T__ (Table). Responses addressing weaknesses are labeled __AR1W#__, and those addressing questions are labeled __AR1Q#__.
>
> While we aim to address all concerns as thoroughly as possible within this rebuttal, some questions may be clarified further during the discussion period. We appreciate your patience and engagement.
>
> ---
> > __R1W2__: Comparison to more baselines (Isotropic Merging, TSV-M, and KnOTS).
>
> __AR1W2__:  Results are shown in Table __AR1W2T1__.
>
> **Table AR1W2T1: Comparison of Different Merging Methods on Eight Vision Merging Tasks**
> | Method          | ViT-B-32 | ViT-L-14 |
> |----------------|----------|----------|
> | TSV-M           | 85.5     | 93.0    |
> | Isotropic      | 86.2     | 94.7     |
> | RegMean++      | 86.2     | 91.2    |
> | StatsMerging+Knots  | 82.4    | 90.4    |
> | **StatsMerging**   | **84.5**     | **92.1**     |
> | **StatsMerging++** | **94.5**     | **94.1**     |
>
> *Caption: Comparison of merging and baseline methods evaluated on ViT-B-32 and ViT-L-14 models (Avg Acc %).*
>
> Overall, StatsMerging++ with fewer unlabelled samples (without labels) was able to outperform RegMean++ and on-par with TSV-M. These results will be added to the revised draft, and Isotropic Merging and TSV-M will be properly cited and discussed. We thank R1 for suggesting this valuable comparison.
>
> We would also like to mention that we already included __Knots__ in the __original submission__, specifically inTable 15 in Line 1471 in _A.5 EXTENDED RELATED WORK, Relationship to KnOTS_. We quote the Table here for your convenience. Note the setup differs from Table AR1W2T1 so the accuracies are different.
>
> **Table 15: Comparison of Different Merging Methods on Two-Task Merging (RE, EU)**
> | Method | RE | EU | Avg Acc (%) |
> |-|-|-|-|
> | Individual             | 96.1  | 99.7  | 97.9             |
> | Task Arithmetic        | 85.2  | 96.7  | 90.9             |
> | TIES-Merging           | 86.4  | 97.2  | 91.8             |
> | *StatsMerging* + KnOTS | 92.1  | 94.2  | 93.2             |
> | **_StatsMerging_**     | **96.0** | **98.0** | **97.4 (+4.2)** |
>
> ---
>
> > __R1W3__: Task-level StatsMerging vs layer-wise StatsMerging
>
> __AR1W3__: We intentionally keep the main results in Fig. 4 concise using only LW StatsMerging to represent our method while keeping the extended details in the Appendix. This presentation setup (one main LW result in the main paper while keeping TW vs LW results in the Appendix) also aligns with the experimental result structure of the prior work AdaMerging [5]. In the __original submission__, we have already clarified and explicitly mentioned **task-level** results with an explicit reference to the TW details on Page 7. We quote the __original description__ below for your convenience:
>
> _"Please refer to the Appendix for experimental details, including the full list of tasks, datasets, baselines,
> along with the_ __task-level__ _results in Sections A.1 and A.2, respectively."_
>
> We list below the figure and table numbers containing the the (TW) StatsMerging results in the __original submission__ for your reference: Figure 7, Table 3, Table 7 & Table 14. We quote Table 3 in _Section A.2 DETAILS OF TASK-LEVEL RESULTS_, which compares TW vs. LW StatsMerging alongside AdaMerging.
>
> **Table 3: Multi-task merging performance (Avg Acc %) when merging ViT-B/32 models on eight vision tasks**
>
> | Method              |  SU  |  CA  |  RE  |  EU  |  SV  |  GT  |  MN  |  DT  | Avg Acc |
> |-|-|-|-|-|-|-|-|-|-|
> | TW AdaMerging       | 58.0 | 53.2 | 68.8 | 85.7 | 81.1 | 84.4 | 92.4 | 44.8 | 71.1 |
> | TW AdaMerging++     | 60.8 | 56.9 | 73.1 | 83.4 | 87.3 | 82.4 | 95.7 | 50.1 | 73.7 |
> | **TW StatsMerging** | 61.3 | 70.0 | 74.2 | 85.2 | 87.5 | 82.5 | 96.2 | 54.2 | 76.4 |
> | LW AdaMerging       | 64.5 | 68.1 | 79.2 | 93.8 | 87.0 | 91.9 | 97.5 | 59.1 | 80.1 |
> | LW AdaMerging++     | 66.6 | 68.3 | 82.2 | 94.2 | 89.6 | 89.0 | 98.3 | 60.6 | 81.1 |
> | **LW StatsMerging** | 67.4 | 74.1 | 82.9 | 91.1 | 89.8 | 94.7 | 98.3 | 77.5 | 84.5 |
> | **LW StatsMerging++** | **92.4** | **95.4** | **95.1** | **92.9** | **94.6** | **98.7** | **98.5** | **88.4** | **94.5 (+5.1)** |
>
> [5] Enneng Yang, Zhenyi Wang, Li Shen, Shiwei Liu, Guibing Guo, Xingwei Wang, and Dacheng Tao.
> Adamerging: Adaptive model merging for multi-task learning. arXiv preprint arXiv:2310.02575,
> 2023.

---

> ### Author Response · Authors · 2025-11-20
> **Response to Reviewer ksSD (R1) - Questions**
>
> > __R1Q2.1__: Data sources: What exact data are used for training StatsMergeLearner and for distillation?
>
> __AR1Q2.1__: The __unlabeled training samples__ $x$ and the __pseudo-labels__ $\hat{y}$ produced by the task-specific teachers $\theta$, as already described in the __main paper__ in the __original submission__ on Page 6, Line 293, formally described in the loss function in Equation 7 (using __only pseudo-label__ $\hat{y}$ without any ground truth $y$), and illustrated in Figure 3 (_“Depiction of Task-Specific Teacher Distillation procedure”_) as well as Algorithm 1 (_Unified Statistics-Guided Model Merging via Task-Specific Teacher Model Distillation_). We also provide an explicit pointer to the extended description in the text on Page 6: “Distillation is detailed in Appendix A.3.”
>
> We quote the __original descriptions__ for your convenience:
>
> Line 293 Page 6: "the predictions $\hat{y}_{i,k}$ from the model trained on task $k$ are sufficiently reliable to serve as high-quality pseudo-labels for the corresponding pre-trained dataset sample {$x_i$, $y_i$}_k. We aggregate such pairs {$x_i$, $\hat{y}_i$}_k to construct the merged dataset to train SML."
>
> **A.3 Details of Task-Specific Teacher Distillation**
>
> 1. **Task-Specific Teacher Models Preparation.**
>    Collect $K$ pre-trained models
>    $\Theta$ = {$\theta_1, \theta_2, \ldots, \theta_k$\},
>    where each model weight is fine-tuned on an independent task $k$ with dataset
>    {$x_i, y_i$}_k $\in D_k$.
>    $D_k$ denotes the dataset for task $k$, $x_i$ and $y_i$ represent a sample’s input and its corresponding label.
>    Note that $y_i$ is not used for SML learning but only in the evaluation step.
>
> 2. **Train/Val/Test Split.**
>    Each dataset $D_k$ for task $k$ is split into training, validation, and test sets with an 8:1:1 ratio, denoted as
>    $D_k^{\text{train}}, D_k^{\text{val}}, and D_k^{\text{test}}$, respectively.
>
> 3. **Pseudo Label Preparation for Training Set $D^{\text{train}}$.**
>    Following (2), for task $k$, the task-specific teacher $\theta_k$ takes a sample $x_{i,k}$ and generates its prediction $\hat{y}_{i,k}$ as a pseudo label.
>
>    The resulting pairs $(x_{i,k}, \hat{y}_{i,k})$ are aggregated to form task $k$’s training dataset
>    $D_k^{\text{train}} \subseteq D^{\text{train}}$.
>
> 4. **Val $D^{\text{val}}$ and Test $D^{\text{test}}$ Preparation.**
>    Following (2), for task $k$, the original pairs $(x_{i,k}, y_{i,k})$ in the split validation set $(D_k^{\text{val}} \subseteq D^{\text{val}})$
>    or test set $(D_k^{\text{test}} \subseteq D^{\text{test}})$ are used, where $y_{i,k}$ is the human-annotated ground truth label used solely for evaluation.
>
> 5. **Complete Dataset $D$ Preparation.**
>    Aggregate $D^{\text{train}}, D^{\text{val}}, and D^{\text{test}}$ to form the complete dataset D = {$D^{\text{train}}, D^{\text{val}}, D^{\text{test}}$}.
>
>    Concretely, in the eight vision tasks, the samples $\{x_i, y_i\}_k \in D_k$ are drawn from the following datasets:
>    **SUN397 (SU), Cars (CA), RESISC45 (RE), EuroSAT (EU), SVHN (SV), GTSRB (GT), MNIST (MN), and DTD (DT).**
>
>    The pseudo label $\hat{y}_i$ is generated by the task-specific teacher set $\Theta =$ {$\theta_1, \theta_2, \ldots, \theta_k$}.
>    These are aggregated to constitute the overall dataset $D$. The same procedure applies to the NLP tasks.
>
> ---
>
> > __R1Q2.2__: Is it the training samples from each task?
>
> __AR1Q2.2__: __Yes__, as described in Line 293 on Page 6 with extended details in __AR1Q2.1__ 3. **Pseudo Label Preparation for Training Set $D^{\text{train}}$.**  and in Line 1084 in the __original submission__. We quote the __original description__ for your convenience:
>
> "Following (2), for __task__ $k$, the task-specific teacher $\theta_k$ takes a __sample__ $x_{i,k}$ and generates its prediction $\hat{y}_{i,k}$ as a pseudo label.
>
> The resulting pairs $(x_{i,k}, \hat{y}_{i,k})$
> are aggregated to form task $k$’s training dataset
> $D_k^{\text{train}} \subseteq D^{\text{train}}$."
>
> >__R1Q2.3__: Also, is the setup strictly unlabeled for both training and validation, or is any labelled data employed?
>
> __AR1Q2.3__: __Correct__. __No labelled data__ for training and validation. __Labels__ are used purely for evaluation.
> From the __original submission__, these are explicitly described on Page 6, Line 293, in the loss function in Equation 7 (using __only pseudo-label__ $\hat{y}$ without any ground truth $y$), and illustrated in Figure 3, detailed on Line 1085 1. Task-Specific Teacher Models Preparation:
>
> "Note that $y_{i}$ is not used for SML learning but only in the evaluation step."
>
> and Line 1094 in 4. Val $D^{\text{val}}$ and Test $D^{\text{test}}$ Preparation:
>
> "$y_{i,k}$ is the human-annotated ground truth label used solely for evaluation."

---

> ### Author Response · Authors · 2025-11-20
> **Response to Reviewer ksSD (R1) - Questions**
>
> > __R1Q3.1__: Training cost: Line 332 states StatsMergeLearner (SML) is trained for 500 epochs, which seems heavy compared to, e.g., Adamerging’s ~500 steps
>
> __AR1Q3.1__: It would be great if you further clarify the differences between (Q3.1.1 StatsMergeLearner for 500 epochs) and (Q3.1.2 Adamerging’s ~500 steps). I assume by "steps" you meant the same thing as "epochs"? It is unclear why the question asks about Q3.1.1 StatsMergeLearner being heavier. It would be appreciated if you could clarify that.
>
> We claim SML lightweight due to its __simplicity__ (simple MLP is sufficient) and the fact that it __only introduces marginal parameter and computation overhead__ (only __3.06%__ more #params and __0.05%__ more FLOPs) on top of the merged model. During training, only these small amount of parameters in MPL are updated. Therefore the training cost remains on par with AdaMerging. We quote the __original description__ on Line 382 in the __original submission__ for your reference:
>
> _"Without the merged model, StatsMergeLearner (SML) itself is orders of magnitude smaller and computationally lighter than the merged model, with only 0.336M parameters, 0.73M MACs and 1.46M FLOPs. The results demonstrate that SML introduces negligible overhead in terms of parameters (SML-to-Merged Model Parameter Ratio: 0.336M / 10.99M = 0.0306) and computation (SML-to-Merged Model Compute Ratio: 1.46M / 2.95G = 0.0005)."_
>
> > __R1Q3.2__: Do (a) the number of models being merged and (b) backbone size (e.g., ViT-L/14) influence the number of training steps used to train the StatsMergeLearner?
>
> __AR1Q3.2__: We observed that the converged point at the specific training epoch differ for (a) and (b). We mainly report the performance in the final SML trained for 500 epochs as the main results.

---

> ### Author Response · Authors · 2025-11-21
> **Response to Reviewer ksSD (R1) - Questions**
>
> > __R1Q1__: Effect of extra validation data: The jump from LW-StatsMerging to LW-StatsMerging++ is substantial.
>
> >__R1Q1.1__: Do other adaptation methods (e.g., Adamerging) show similar improvements as validation data increases?
>
> __AR1Q1.1__: __Yes__, as shown in __Table AR1Q1.1T1__. However, the performance gain of StatsMerging++ with more validation data is much larger than that of AdaMerging++. When the data rate increases from 1% to 5%, LW StatsMerging++ improves from 84.5% to 94.5%, whereas LW AdaMerging++ only increases from 80.6% to 81.0%. This demonstrates LW StatsMerging++ is more data efficient than LW AdaMerging++.
>
> **Table AR1Q1.1T1 Impact of the amount of available data on performance (Avg Acc %) when merging ViT-B/32 models**
> | **Method**          |  Data | SU  |  CA  |  RE  |  EU  |  SV  |  GT  |  MN  |  DT  | Avg Acc |
> |---------------------|-|------|------|------|------|------|------|------|------|---------|
> | LW AdaMerging       | 1% | 61.9 | 66.3 | 81.8 | 86.0 | 88.6 | 85.8 | 97.4 | 52.5 | 77.5 |
> | LW AdaMerging++     | 1% | 66.9 | 68.6 | 81.4 | 91.8 | 89.2 | 87.1 | 98.1 | 61.8 | 80.6 |
> | **LW StatsMerging** | 1% | 67.4 | 74.1 | 82.9 | 91.1 | 89.8 | 94.7 | 98.3 | 77.5 | 84.5 |
> |
> | LW AdaMerging       | 5% | 63.7 | 68.6 | 79.1 | 93.3 | 86.5 | 91.7 | 97.2 | 61.9 | 80.1 |
> | LW AdaMerging++     | 5% | 66.4 | 68.4 | 81.5 | 92.9 | 90.0 | 89.0 | 98.2 | 61.5 | 81.0 |
> | **LW StatsMerging++** | 5% | **92.4** | **95.4** | **95.1** | 92.9 | **94.6** | 98.7 | **98.5** | **88.4** | **94.5 (+5.1)** |
> |
> | LW AdaMerging       | 100% | 64.5 | 68.1 | 79.2 | 93.8 | 87.0 | 91.9 | 97.5 | 59.1 | 80.1    |
> | LW AdaMerging++     | 100% | 66.6 | 68.3 | 82.2 | 94.2 | 89.6 | 89.0 | 98.3 | 60.6 | 81.1    |
>
> __Difference between StatsMerging++ and AdaMerging++__:
>
> _StatsMerging++_: 5% validation data is used instead of 1% in StatsMerging.
>
> _AdaMerging++_: additionally removes parameter redundant values and sign conflicts using Ties-Merging. Not related to the amount of data.
>
> We will add an additional description and Table AR1Q1.1T1 in the updated the draft to enhance this clarity. We thank Reviewer ksSD (R1) for the valuable feedback, which helped make our experimental evaluation more thorough and comprehensive.
>
> Note that in Table 3 in the original submission, we reported the results of AdaMerging and AdaMerging++ using 100% data, i.e. 80.1% and 81.1%, respectively.
>
> ---
>
> > __R1Q1.2__: Additionally, since StatsMerging does not introduce any new hyperparameters of its own, what is the role of validation data?
>
> __AR1Q1.2__: __Role__: it provides additional information/signals of a task (without labels) to guide merging.
>
> More validation data provides stronger guidance for merging because it yields a more accurate estimation of task data distributions and allows more pseudo-labels to be generated by the task-specific teachers. Together with more reliable weight-statistics distributions, this richer supervision helps SML, implemented as a simple 2-layer MLP, better determine task importance and compute more effective merging coefficients.
>
> We quote the text in the __original submission__ to address this question:
> _"We attribute the improvements to the ability of StatsMergeLearner to adapt task-
> specific weights based on their weight statistics to the merged model. The use of pseudo labels
> from task-specific teachers provides stronger signals for StatsMergeLearner in assigning weight
> coefficients compared to AdaMerging entropy minimization and more complex task-adaptive expert
> selection mechanism in WEMoE."_

---

> ### Author Response · Authors · 2025-12-03
> **Response to Reviewer ksSD (R1) - Weaknesses**
>
> > __R1W1__: Resultsin 14- and 20-task benchmarks on various backbone sizes (ViT-B/32 and ViT-L/14).
>
> __AR1W1__:  Results are shown in Table __AR1W1T1__.
>
>
> **Table AR1W1T1: Comparison of Different Merging Methods on the Vision Merging Benchmark (8, 14 and 20 Tasks) with ViT-B/32 and ViT-L/14 Backbones**
> | Method                 | Backbone | 8 Tasks | 14 Tasks | 20 Tasks |
> |-----------------------|-|---------|----------|----------|
> | Pre-Trained           | ViT-B/32 | 48.4    | 57.3     | 56.1     |
> | Weight Averaging      | ViT-B/32 | 66.5    | 64.4     | 61.1     |
> | Task Arithmetic       | ViT-B/32 | 70.8    | 65.4     | 60.6     |
> | TIES-Merging                | ViT-B/32 | 75.1    | 68.0       | 63.4     |
> | RegMean++             | ViT-B/32 | 84.4    | –        | 77.0     |
> | **StatsMerging++ (Ours)** | ViT-B/32 | **94.5** | **90.7** | **86.8** |
> | Pre-Trained           | ViT-L/14 | 64.4    | 68.0      | 65.1     |
> | Weight Averaging      | ViT-L/14 | 79.4    | 76.6     | 71.5     |
> | Task Arithmetic       | ViT-L/14 | 84.8    | 79.3     | 74.0     |
> | TIES-Merging                 | ViT-L/14 | 86.9    | 79.5     | 75.7     |
> | RegMean++             | ViT-L/14 | 91.2   | –        | 81.0       |
> | **StatsMerging++ (Ours)** | ViT-L/14 | **94.1** | **89.1** | **88.9** |
>
> Note RegMean++ does not report its performance for the 14 tasks in its original paper [8].
>
> [8] Nguyen, T.H., Huu-Tien, D., Suzuki, T. and Nguyen, L.M., 2025. RegMean++: Enhancing Effectiveness and Generalization of Regression Mean for Model Merging. arXiv preprint arXiv:2508.03121.

---

### Author Response · Authors · 2025-11-25
**Follow-Up on Posted Responses**

Dear Reviewer ksSD (R1), 5Rvs (R2) and kzYd (R4), we have posted responses to several of your questions while we continue working on the remaining ones. It would be greatly appreciated if you could let us know whether our current answers address your concerns or if you have any further questions. Thank you.

---

### Author Response · Authors · 2025-12-03
**Complete and Thorough Rebuttal: Responses to All Reviewer Questions and Weaknesses**

We thank the reviewers for their time and constructive feedback. We have provided __COMPLETE__ responses to __ALL__ identified weaknesses and questions. Thank you.

---

### Author Response · Authors · 2025-12-03
**Author Final Remarks**

U: Updated. A: Answer. R: Reviewer. W: Weakness. F: Follow-Up. T: Table. Sec: Section in the updated draft.

We appreciate the reviewers’ constructive feedback and the opportunity to clarify our contributions. We believe the revisions, which have been highlighted in tear in the __updated draft__, and responses __address ALL concerns__, strengthen the technical soundness of the work, and provide clearer empirical and theoretical justification. We hope the reviewers agree that the paper offers a meaningful and timely contribution to model merging, and we thank them again for their time and thoughtful engagement.

We summarize the main remarks addressed in our rebuttal as follows:
- **Scaling Tasks**: Included 8, 14 to 20 tasks. (__AR1W1, Sec 4.2 T3__)
- **More Baselines**: Added Isotropic Merging, TSV-M, RegMean++ and KnOTS. (__AR1W2, AR4W2, Sec 4.2 Fig. 4__)
- **Effect of Data Size**: (__AR1Q1, AR3W2, AR3Q2, Sec A.5.6 T15, T16__)
- **Weight Distribution Design Choice**: Justified using mean, variance, magnitude, and SVD Rank-3 as key efficient statistics.
  - (**Empirical Evidence**: __AR2W1, AR3W1, AR3Q1, AR4W4, Sec 4.3 T5__)
  - (**Theoretical Evidence**: __AR4W5, Sec 3.2 Motivation, Sec A.4__)

---

### Meta-Review · Area_Chair_4NQ9 · 2026-01-06

**Summary:**

This paper proposes StatsMerging, a lightweight learning-based model merging method designed to adaptively predict merging coefficients to consolidate task-specific models into a single one. The core of the approach is a network using weight distribution statistics to serve as proxies for task or layer importance. To obviate the need for human annotations, the authors propose to use task-specific distillation in which   teacher models generate pseudo-labels to guide training. The paper further introduces a "distill-then-merge" paradigm to implement merging for architecturally heterogeneous models. Experimental results on merging eight vision and seven NLP tasks show that StatsMerging outperforms baselines like AdaMerging and WEMoE.

The main issues raised by reviewers that contribute to the Reject recommendation are:

+ **Fairness, clarity, and data usage (ksSD, 5Rvs, RxsX)**: Multiple reviewers questioned the fairness and precise details of the use of training data in the proposed approach. The author rebuttal did not clarify in any significant way how much training data from the original tasks is actually used. Moreover, the precise definition and usage of "validation" also remains unclear from the details of the original manuscript and clarifications in rebuttal. Given that the bulk of comparisons are made with methods requiring no training data, this is a serious weakness of the work in its current form.
+ **Missing methods from comparative analysis (kzYd, ksSD)**: The most recent data-free approaches (Iso-C, Iso-CTS, and TSV) were not included in the original experiments, nor were results on merging more tasks and using a broader range of base models. These were added in rebuttal, however this further narrows the gap between the (expensive) training-based StatsMerging approach and the cheaper, data-free state-of-the-art (except on very small models like ViT-B/32).
+ **Efficiency (ksSD)**: One reviewer raised issues related to efficiency, given that StatsMerging requires 500 epochs of training on (all?) training data from the original tasks, which could potentially incur massive extra costs especially in light of the already good performance of training-free approaches.
+ **Novelty and impact (5Rvs)**: Recent approaches based on the statistical properties of weight matrices (TSV, Iso-C, and Iso-CTS) have already established the utility of statistical (mean, variance) and algebraic (SVD) properties to perform merging. Thus, it is unsurprising that these elements are useful for determining mixing coefficients. The added benefit of knowledge distillation is unsurprising, especially given that the improvement is much more marked for the small, low-capacity ViT-B/32 model.

Reviewers seem to agree that there are interesting ideas in this paper, however problems with clarity, novelty, and potential impact outweigh the strong points and the recommendation is to Reject.

**Reviewer Concerns:**

The bulk of the rebuttal and subsequent revision was dedicated to addressing many issues related to missing approaches, model architectures, and merging of more tasks in order to bring the work into alignment with the recent literature. The issue with the *missing* approaches and experiments was thus resolved, however this in my opinion already constitutes a major revision of the work and thus should be considered separately. Moreover, the new results -- especially on larger models -- narrows the gap between the proposed approach and the recent, training-free state-of-the-art and, essentially, raises even more questions than they resolve.

Issues related to use of training data from the original task datasets were not adequately addressed in rebuttal, nor were questions regarding efficiency or novelty of the contribution.

**Reviewer Scores:**

+ **R1 (ksSD)**: The most articulated review which raised a broad range of issues with the original manuscript. The author rebuttal regarding missing methods and architectures might have convinced this reviewer, but I do not believe their concerns regarding details data usage would have been assuaged by the rebuttal.
+ **R2 (5Rvs)**: A short review with little substantive content. Did not factor significantly into the final recommendation, aside from drawing attention to potential problems with novelty, impact, and re-emphasizing the lack of clarity regarding data usage.
+ **R3 (RxsX)**: Raised questions about theoretical motivations, inconsistencies in experimental results, and (again) details regarding data usage. The only reviewer to engage during the limited discussion period, however were not satisfied with author rebuttal regarding data usage.
+ **R4 (kzYd)**: Also commented that there were training-free methods missing from the comparative analysis and noted some odd aspects in training behavior. They might have been convinced by the inclusion of training-free methods, however as stated above these additions are so significant as to consider the manuscript substantially different than the original submission.

---

### Decision · Program_Chairs · 2026-01-26

Reject